# TOWARD EFFICIENT EXPLORATION
# BY LARGE LANGUAGE MODEL AGENTS

**Dilip Arumugam**
Department of Computer Science
Princeton University
`dilip.a@cs.princeton.edu`

**Thomas L. Griffiths**
Department of Computer Science
Department of Psychology
Princeton University
`tomg@princeton.edu`

## ABSTRACT

A burgeoning area within reinforcement learning (RL) is the design of sequential decision-making agents centered around large language models (LLMs). While autonomous decision-making agents powered by modern LLMs could facilitate numerous real-world applications, such successes demand agents that are capable of data-efficient RL. One key obstacle to achieving data efficiency in RL is exploration, a challenge that we demonstrate many recent proposals for LLM agent designs struggle to contend with. Meanwhile, classic algorithms from the RL literature known to gracefully address exploration require technical machinery that can be challenging to operationalize in purely natural language settings. In this work, rather than relying on finetuning or in-context learning to coax LLMs into implicitly imitating a RL algorithm, we illustrate how LLMs can be used to explicitly implement an existing RL algorithm (Posterior Sampling for Reinforcement Learning) whose capacity for statistically-efficient exploration is already well-studied. We offer empirical results demonstrating how our LLM-based implementation of a known, data-efficient RL algorithm can be considerably more effective in natural language tasks that demand prudent exploration.

## 1 INTRODUCTION

Large language models (LLMs) have rapidly permeated many areas of machine learning, demonstrating proficiency across a broad range of tasks (Bommasani et al., 2021; Achiam et al., 2023; Touvron et al., 2023; Team et al., 2023; Hurst et al., 2024; Jaech et al., 2024). This has inspired recent work studying how LLMs can best be used to solve sequential decision-making problems (Silver & Sutton, 2025). These efforts have led to the introduction of new designs for LLM agents that aim to learn optimal behavior through trial-and-error interaction within natural language environments (Yao et al., 2023; Shinn et al., 2024; Monea et al., 2024; Klissarov et al., 2025). While details vary by approach, broadly speaking these new agent designs involve one or more LLMs that interact to ultimately select actions within the environment. However, such agents still reside in the classic RL setting (Sutton & Barto, 1998) and, consequently, must still grapple with the fundamental obstacles to data efficiency (generalization, exploration, and credit assignment) that the RL literature has studied for decades.

While composing LLMs to arrive at new agent designs is the current norm, we propose that an alternative strategy is to re-examine existing RL algorithms and consider how LLMs might implement them in otherwise inaccessible environments. An RL algorithm consists of specifying inputs and detailing a sequence of steps for determining behavior at each time period. Why should the emergence and proliferation of LLMs change the fundamental principles of agent design? Instead, as visualized in Figure 1, perhaps LLMs can be used to create new, potentially-inexact incarnations of existing RL algorithms via the subroutines needed to implement them.

In this work, we focus on data-efficient RL with LLMs and isolate the key challenge of exploration. We demonstrate how modern LLMs afford a contemporary implementation of an existing RL algorithm, Posterior Sampling for Reinforcement Learning (PSRL) (Strens, 2000; Osband et al., 2013), that is both well-studied and whose capacity for good exploration is already known to yield provably-efficient RL in a number of problem classes. We empirically find that our LLM-based

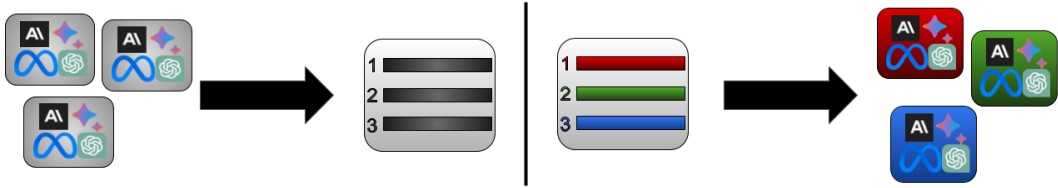

Figure 1: Abstractly, an RL algorithm is an ordered sequence of steps. Existing approaches for LLM agent design (left) orchestrate some number of LLMs to implicitly induce a RL algorithm. In contrast, this paper advocates for a novel agent design principle (right) whereby an existing RL algorithm is explicitly implemented by outsourcing individual steps to distinct LLMs.

implementation of PSRL retains the strong exploration properties that, up to this point, have not only been primarily restricted to tabular domains but also been absent in recent designs for LLM agents. We further observe that the choice of LLM underlying the PSRL implementation matters and, in an environment with stochastic transition dynamics, show that upgrading to a more capable model (GPT-4o to o1-mini) is the difference between incurring linear regret and obtaining cumulative regret on par with classic PSRL. Altogether, our work underscores the importance of addressing exploration in the design of LLM agents, illustrates the considerable value that decades of RL research have to offer data-efficient decision-making with LLMs, and establishes a key distinction between LLMs that implement a RL algorithm versus a RL algorithm that is implemented with LLMs.

## 2 PROBLEM FORMULATION

All random variables are defined on a probability space $(\Omega, \mathcal{F}, \mathbb{P})$. For any arbitrary set $\mathcal{X}$, we use $\Delta(\mathcal{X})$ to denote the set of all probability distributions with support on $\mathcal{X}$. For any $N \in \mathbb{N}$, we denote the index set as $[N] = \{1, 2, \ldots, N\}$.

We formulate a sequential decision-making problem as a finite-horizon, episodic Markov Decision Process (MDP) (Bellman, 1957; Puterman, 1994) defined by $\mathcal{M} = \langle \mathcal{S}, \mathcal{A}, \mathcal{R}, \mathcal{T}, \beta, H \rangle$. $\mathcal{S}$ is a set of states, $\mathcal{A}$ is a set of actions, $\mathcal{R} : \mathcal{S} \times \mathcal{A} \to [0, 1]$ is a reward function providing evaluative feedback in the unit interval, $\mathcal{T} : \mathcal{S} \times \mathcal{A} \to \Delta(\mathcal{S})$ is a transition function prescribing distributions over next states, $\beta \in \Delta(\mathcal{S})$ is an initial state distribution, and $H \in \mathbb{N}$ is the maximum episode length or horizon. Within each of $K \in \mathbb{N}$ total episodes, the agent acts for $H$ steps beginning with an initial state $s_1 \sim \beta(\cdot)$ and, at each timestep $h \in [H]$, observes the current state $s_h \in \mathcal{S}$, selects an action $a_h \in \mathcal{A}$, enjoys a reward $r_h = \mathcal{R}(s_h, a_h)$, and transitions to a next state $s_{h+1} \sim \mathcal{T}(\cdot \mid s_h, a_h)$.

An agent is characterized by its non-stationary, stochastic policy $\pi : \mathcal{S} \times [H] \to \Delta(\mathcal{A})$, which encodes a pattern of behavior by mapping individual states and the current timestep to a probability distribution over actions. We assess the performance of a policy $\pi$ in MDP $\mathcal{M}$ at timestep $h \in [H]$ when starting at state $s \in \mathcal{S}$ and taking action $a \in \mathcal{A}$ by its associated action-value function $Q_{\mathcal{M},h}^\pi(s, a) = \mathbb{E}\left[\sum_{h'=h}^{H} \mathcal{R}(s_{h'}, a_{h'}) \mid s_h = s, a_h = a\right]$. Taking the value function as $V_{\mathcal{M},h}^\pi(s) = \mathbb{E}_{a \sim \pi_h(\cdot \mid s)}\left[Q_{\mathcal{M},h}^\pi(s, a)\right]$, we define the optimal policy $\pi^\star$ as achieving supremal value $V_{\mathcal{M},h}^\star(s) = \sup_{\pi \in \Pi} V_{\mathcal{M},h}^\pi(s)$ for all $s \in \mathcal{S}$, $h \in [H]$ where $\Pi$ denotes the class of all non-stationary, stochastic policies. For any episode $k \in [K]$, we let $\tau_k = (s_1^{(k)}, a_1^{(k)}, r_1^{(k)}, \ldots, s_H^{(k)}, a_H^{(k)}, r_H^{(k)}, s_{H+1}^{(k)})$ denote the random trajectory experienced by the agent executing its policy in the environment. Meanwhile, $H_k = \{\tau_1, \tau_2, \ldots, \tau_{k-1}\} \in \mathcal{H}$ is the entire random history of interaction at the $k$th episode.

Abstractly, a RL algorithm is a sequence $\{\pi^{(k)}\}_{k \in [K]}$ where the policy deployed at each episode $\pi^{(k)}$ is a function of the current history $H_k$. We may evaluate the performance of a RL algorithm on MDP $\mathcal{M}$ via its cumulative regret: $\text{REGRET}(\{\pi^{(k)}\}_{k \in [K]}, \mathcal{M}) = \mathbb{E}\left[\sum_{k=1}^{K} \left(V_{\mathcal{M},1}^\star(s_1) - V_{\mathcal{M},1}^{\pi^{(k)}}(s_1)\right) \mid \mathcal{M}\right]$, which aggregates performance shortfall between an agent's chosen policy and the optimal policy in all episodes. Naturally, an agent designer seeks out a RL algorithm with minimal cumulative regret.

# 3 LLM IMPLEMENTATION OF POSTERIOR SAMPLING FOR REINFORCEMENT LEARNING

One of the major obstacles to data-efficient RL is exploration, where a learner must determine what data to collect from the environment to maximize long-term performance. While much of the early work on addressing exploration in RL (see Appendix A for a detailed review of prior work) adhered to "optimism in the face of uncertainty," an alternative is to proceed in a Bayesian fashion.

The Bayesian RL setting (Bellman & Kalaba, 1959; Duff, 2002; Ghavamzadeh et al., 2015) recognizes that the underlying MDP $\mathcal{M}$ is entirely unknown to the agent and, therefore, a random variable. The agent is thus endowed with a prior distribution $\mathbb{P}(\mathcal{M})$ to reflect initial uncertainty in the true MDP. While the standard RL objective (Sutton & Barto, 1998) calls for an agent to minimize regret, another performance criterion is the Bayesian regret, which simply integrates out the randomness in $\mathcal{M}$ with respect to an agent's prior: $\text{BAYESREGRET}(\{\pi^{(k)}\}_{k\in[K]}) = \mathbb{E}\left[\text{REGRET}(\{\pi^{(k)}\}_{k\in[K]}, \mathcal{M})\right]$. We make a standard assumption that the prior is well-specified and the true MDP resides in its support.

Unfortunately, the canonical Bayes-Adaptive MDP (BAMDP) (Bellman & Kalaba, 1959; Duff, 2002) that encapsulates the full Bayesian RL problem is often computationally-intractable even in the simplest classes of environments with precious few exceptions (Gittins, 1979). This is a direct consequence of the intractably-large BAMDP hyperstate space (Duff, 2002; Arumugam & Singh, 2022), in which traditional MDP states are folded in alongside *epistemic states* (Lu et al., 2023) that contain an agent's beliefs and epistemic uncertainty (Der Kiureghian & Ditlevsen, 2009) about the world. The MDP transition and reward functions are unknown to a RL agent and, with each step taken in the true environment, the resulting reward and next-state transition provide ground-truth observations by which the agent may refine posterior beliefs about the underlying MDP $\mathcal{M}$. Even for a simple finite MDP, the epistemic state space is exponentially-large in the problem horizon $H$. One might hope that the epistemic state could be lazily updated while still enabling strategic exploration by reducing epistemic uncertainty; this insight is the basis of posterior-sampling methods in RL.

## 3.1 THE CLASSIC APPROACH

The promise of Bayesian RL methods is to facilitate statistically-efficient exploration by reducing an agent's epistemic uncertainty about the world. One strategy for reaping the benefits of uncertainty-based exploration in a computationally-tractable manner is through Posterior Sampling for RL (PSRL) (Strens, 2000), presented as Algorithm 1. Rather than updating the epistemic state at each timestep, PSRL holds it fixed during each episode and only updates the posterior at the end using the full trajectory $\tau_k$. To govern action selection within each episode based on current knowledge of the true underlying MDP $\mathbb{P}(\mathcal{M} \mid H_k)$, PSRL employs Thompson sampling (TS) (Thompson, 1933; Russo & Van Roy, 2014; 2016; Russo et al., 2018), whereby the agent draws one posterior sample as a statistically-plausible hypothesis about the true MDP (Line 3) and proceeds to act optimally with respect to it by executing the sampled MDP optimal policy (Lines 4-5). It has been shown theoretically that, by iteratively employing TS in this manner, PSRL is able to achieve strong exploration and satisfy Bayesian regret upper bounds for statistically-efficient RL in tabular MDPs and beyond (Osband et al., 2013; Osband & Van Roy, 2014; Abbasi-Yadkori & Szepesvari, 2014; Osband & Van Roy, 2016; Agrawal & Jia, 2017; Ouyang et al., 2017; Osband & Van Roy, 2017; Lu & Van Roy, 2019; Arumugam & Van Roy, 2022; Xu et al., 2024). A key contribution of this work is expanding empirical support for PSRL, an algorithm that has largely been a method of theoretical study up to this point.

While PSRL enjoys nice theoretical guarantees, practical implementations extending beyond tabular MDPs (Osband et al., 2013) face significant computational hurdles. Representing and maintaining epistemic uncertainty about the underlying MDP transition and reward functions is an open challenge in high-dimensional environments. While some work has studied using neural networks to address the broader problem of uncertainty estimation for guiding exploration in RL (Osband et al., 2016a; Lu & Van Roy, 2017; Osband et al., 2018; O'Donoghue et al., 2018; Dwaracherla et al., 2020; Osband et al., 2023; Sasso et al., 2023), the overwhelming majority of these efforts have concentrated on a model-free analogue of PSRL that maintains a Bayesian posterior over the optimal action-value function $Q^\star$ (Osband et al., 2016b; 2019) in lieu of the underlying MDP $\mathcal{M}$. Meanwhile, the minority of such methods that actually strive to implement PSRL have either been met with mixed results across hard-exploration problems or have been limited to evaluations in smaller-scale domains.

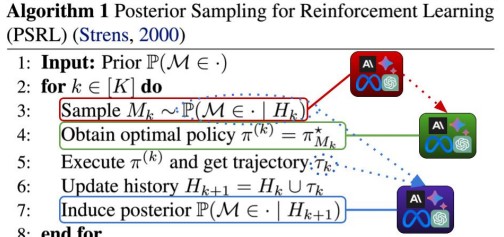

**Algorithm 1** Posterior Sampling for Reinforcement Learning (PSRL) (Strens, 2000)

1: **Input:** Prior $\mathbb{P}(\mathcal{M} \in \cdot)$
2: **for** $k \in [K]$ **do**
3:     Sample $M_k \sim \mathbb{P}(\mathcal{M} \in \cdot \mid H_k)$
4:     Obtain optimal policy $\pi^{(k)} = \pi^\star_{M_k}$
5:     Execute $\pi^{(k)}$ and get trajectory $\tau_k$
6:     Update history $H_{k+1} = H_k \cup \tau_k$
7:     Induce posterior $\mathbb{P}(\mathcal{M} \in \cdot \mid H_{k+1})$
8: **end for**

Figure 2: The PSRL algorithm with LLM subroutines of posterior sampling, optimal behavior with respect to a sample, and posterior updating shown. Dotted arrows show data flow.

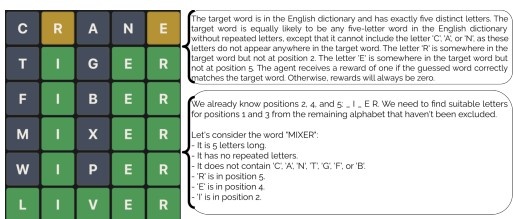

Figure 3: Examples of a posterior (top) and posterior sample (bottom) generated by our LLM-based PSRL in Wordle

Among them is a line of work that leans heavily into the use of Langevin dynamics for recovering the strategic exploration of PSRL (Mazumdar et al., 2020; Karbasi et al., 2023; Ishfaq et al., 2024; Jorge et al., 2024); in the context of this paper, such technical machinery is incredibly challenging and nontrivial to combine or even emulate with LLM agents.

In parallel, beyond the difficulties of maintaining a PSRL agent's posterior distribution over the true MDP, computing the optimal policy for the posterior sample drawn in each episode constitutes an additional challenge that requires solving a planning problem. While there has been progress and even notable successes in this space for deep model-based RL agents (Kaiser et al., 2020), it is unclear if those methods are readily applicable to the natural language tasks faced by LLM agents. In our experiments, while we report positive results for our LLM-based PSRL implementation in MDPs with both deterministic and stochastic transition functions, performance in the latter type of environment eventually deteriorates as the size of the state-action space increases and exacerbates poor LLM planning capabilities under stochastic dynamics (see Appendix D).

## 3.2 A LLM Implementation

The key contribution of this paper is recognizing that LLMs can be operationalized to provide basic, atomic functions from which PSRL may be implemented. This stands in stark contrast to existing strides (see Appendix A) towards efficient decision-making with LLM agents (Nie et al., 2024; Krishnamurthy et al., 2024; Klissarov et al., 2025; Ke et al., 2024), which either leave a LLM to its own devices for strategizing exploration or expect in-context learning (ICL) (Brown et al., 2020) to emulate the exploration of an existing RL or bandit algorithm. While future LLMs may become sufficiently capable to accommodate the former, our experiments today suggest this is not the case for simple, natural-language tasks where efficient exploration is paramount to success; by the same token, we anticipate that our proposed LLM-based implementation of PSRL will also benefit and gracefully extend to more complex natural language tasks as the constituent LLM models become more capable at performing their requested functions. Indeed, we find this to be the case empirically when applying our approach to MDPs with stochastic transition functions. LLM agents emulating the outputs of classic RL methods (Nie et al., 2024) are also bound to the same traditional problem classes whereas LLM-based implementations of RL algorithms may broaden the footprint of those classic algorithms to include natural-language domains that would otherwise be entirely infeasible.

As shown in Algorithm 1, our proposed implementation of PSRL relies on LLMs to play three distinct roles: (1) an approximate posterior updater, (2) a posterior sampler, and (3) an optimal policy with respect to a posterior sample. PSRL requires a prior distribution over MDPs as input and, more generally in any episode, needs a current posterior that accurately reflects the agent's current knowledge *and* uncertainty about the world. For our purposes, such an approximate "posterior"[1] is a textual description that summarizes both the known and uncertain aspects of the true MDP transition and reward function. More importantly, it also explicitly communicates (in some way) the amount of uncertainty an agent has about these aspects of the world. As this textual summary amounts to the PSRL agent's epistemic state representation (Lu et al., 2023), an agent designer may exert

---

[1]For ease of exposition, we will refer to this object as a posterior throughout the remainder of the paper, but acknowledge the distinction between it and the true, statistical object that is the Bayesian posterior distribution.

strong influence over this representation through the verbiage and expression of prior knowledge; as a concrete example, specifying the next-state transition distribution of a tabular MDP in our experiments as a Dirichlet distribution (in language) naturally encourages the LLM-based implementation of PSRL to maintain visitation counts. Of course, an advantage is that agent designers may now leverage the full expressivity and fluidity of natural language for communicating prior knowledge without restriction to the few statistical distributions that afford the computational conveniences of conjugate priors.

Given a current posterior reflecting the agent's knowledge and uncertainty about the world, PSRL must be able to draw one posterior sample from these beliefs. We implement this as a first LLM that, given the agent's current textual posterior (initially set to be the agent designer's input prior) is tasked with generating a plausible hypothesis for how transitions and rewards unfold. In some domains, such as tabular MDPs, it may be natural for this to be an exhaustive list of rewards and next-state transitions for each state-action pair. For more practical scenarios of interest, however, it may be beneficial to prompt this posterior sampling LLM so that it can leverage an environment proxy or lossy surrogate MDP (Lu et al., 2023; Arumugam & Van Roy, 2022) that retains only the salient details needed to determine (near-)optimal behavior. As a concrete example, one of our natural language tasks is the game of Wordle (shown in Figure 3) that, as a MDP, has a transition function and reward function defined entirely around an unknown, five-letter target word. Here, the target word serves as an environment proxy that our LLM-based PSRL agent may directly monitor uncertainty over without meticulously maintaining statistics for rewards and transitions of individual state-action pairs.

With a single posterior sample in hand, a PSRL agent must be able to select actions that would be considered optimal if the sampled MDP truly reflected reality. We implement this as a second LLM tasked with executing actions given the current state that maximize value in a way that is consistent with the natural language hypothesis generated by the posterior sampling LLM. In the simplest case, this optimal sample policy LLM need only be given the posterior sample along with the current state and asked directly to generate an action. In more challenging settings, an agent designer may architect the LLM more carefully via chain-of-thought prompting (Wei et al., 2022; Kojima et al., 2022) to increase the chance of selecting optimal actions consistent with provided hypothesis. Even when this policy is only approximately-optimal with respect to the posterior sample in a given episode, classic PSRL still admits a Bayesian regret bound (see Section 5.4 of Osband (2016a)) and one might hope to see an LLM-based implementation of PSRL empirically exhibit similar robustness in practice.

Upon the completion of an episode with the optimal sample policy LLM acting with respect to the hypothesis of the posterior sampling LLM, we task a third and final LLM with updating the PSRL agent's knowledge and residual uncertainty about the world, akin to an (approximate) posterior update. Given a complete trajectory consisting of reward signals and next-state transitions for exactly $H$ state-action pairs, this posterior LLM must reconcile the agent's prior knowledge at the start of the episode against observed interactions from within the environment. With this last piece of functionality in place, all three LLMs can then be orchestrated to run the PSRL algorithm.

## 4 EXPERIMENTS & DISCUSSION

The goal of our experiments is assessing the extent to which our proposed LLM-based PSRL implementation not only retains the desirable exploration properties that PSRL exhibits empirically within simpler problem domains but also expands the range of problems where these benefits can be realized. To this end, we focus our evaluation on tasks which demand prudent exploration to achieve success and where an agent is minimally encumbered by the orthogonal challenges of generalization and credit assignment. For each task, we present cumulative regret curves (lower, flatter plots indicate better performance) where any shading denotes one standard error. All agents use GPT-4o (Hurst et al., 2024) for their constituent LLMs unless otherwise indicated. We let $\kappa_{\text{sampling}}$, $\kappa_{\pi^\star}$, and $\kappa_{\text{posterior}}$ denote the temperatures of the posterior sampling, optimal sample policy, and posterior update LLMs, respectively. Due to space constraints, we defer further details of our experiments and all prompts used in each task to the Appendix.

For natural language tasks, we compare our LLM-based implementation of PSRL against three baseline LLM agents. In-Context Policy Iteration (ICPI) (Brooks et al., 2023) takes classic policy iteration (Howard, 1960) and offers an implementation via three LLMs, using ICL to elicit a rollout

policy; transition function; and reward function respectively. Together, these models allow for policy improvement via greedy action selection $\pi^{(k)}(s_h) = \arg\max_{a \in \mathcal{A}} Q_{\mathcal{M}}^{\pi^{(k-1)}}(s_h, a)$, with ties broken randomly. In-Context RL (ICRL) (Monea et al., 2024) aims to explore via the stochasticity in LLM responses from sensitivity to the input ICL data. Which episodes are included from a replay buffer for ICL with a LLM policy at each timestep is determined by sampling independent Bernoulli($p$) random variables; we study three distinct values of the keep probability $p \in \{1, 0.5, 0.1\}$. Finally, Reflexion (Shinn et al., 2024) passes each full trajectory through a self-reflection LLM that generates verbal guidance; the total history of verbal guidance is given at each timestep to the LLM policy, along with the current state, for improving the quality of decision-making.

### 4.1 MULTI-ARMED BANDITS

#### 4.1.1 BERNOULLI BANDIT

Following prior work studying the exploratory capabilities of LLMs (Coda-Forno et al., 2023; Binz & Schulz, 2023; Coda-Forno et al., 2024; Krishnamurthy et al., 2024; Nie et al., 2024), we begin the empirical assessment of our LLM-based PSRL with a multi-armed bandit problem (Lai & Robbins, 1985; Bubeck & Cesa-Bianchi, 2012; Lattimore & Szepesvári, 2020). Readers unfamiliar with multi-armed bandits may simply observe them as a special case of a MDP with horizon $H = 1$, singleton state space $|\mathcal{S}| = 1$, and a stochastic (rather than deterministic) reward function. Our evaluation follows that of Krishnamurthy et al. (2024) who chose the simple yet challenging case of a five-armed **Bernoulli bandit** with independent arms and an action gap of $0.2$.[2] The version we evaluate has one randomly-selected optimal arm with rewards drawn from a Bernoulli($0.6$) distribution while all other arms use a Bernoulli($0.4$).

Observe that PSRL specialized to a multi-armed bandit problem mirrors classic TS where, at each timestep, the agent samples one plausible hypothesis for the reward distribution of each arm and then proceeds to select the optimal action believed to achieve highest mean reward under this hypothesis. We compare PSRL implemented with LLMs to classic TS for a Bernoulli bandit with each arm initialized with a Beta($1, 1$) prior. Meanwhile, our LLM-based PSRL agent begins with a prior for each arm specified as a `Beta(1,1)` in natural language. While we fix temperatures $\kappa_{\pi^\star} = \kappa_{\text{posterior}} = 1$, we find that the posterior sampling temperature has profound impact on the performance of our LLM-based PSRL agent. Figure 4 compares TS (run for 1,000 independent trials) against PSRL with four distinct settings of $\kappa_{\text{sampling}}$ (run for 20 independent trials).

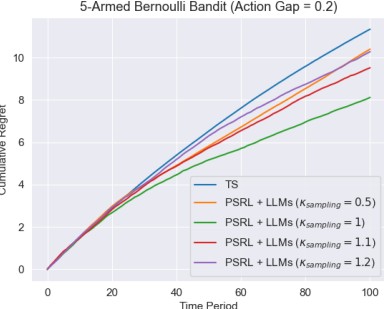

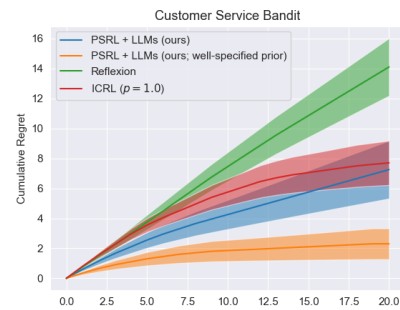

Figure 4: Cumulative regret curves for a 5-armed Bernoulli bandit.

Figure 5: Cumulative regret curves for the real-world customer service bandit.

We find that our LLM-based PSRL achieves a better cumulative regret curve (with $\kappa_{\text{sampling}} = 1.2$) than classic TS, for the limited time horizon of $T = 100$. We find that supplying PSRL with an initial prior of `Beta(1,1)` in language automatically encourages the posterior update LLM to update binary reward observation counts for the chosen arm in each time period. Moreover, we find that the

---

[2] The action gap is defined as the difference in expected reward between the best and second best action. Larger action gaps make it easier to identify the optimal arm with few samples whereas smaller action gaps demand greater exploration.

optimal sample policy LLM has little difficulty in examining the sequence of expected reward values for each arm generated by the posterior sampling LLM and adhering to select the perceived best action. Manipulating $\kappa_{\text{sampling}}$ shows that even values as large as 1 lead to greedy-like exploration in many trials where the resulting posterior sample favors the action observed to yield the most successes thus far. For a limited number of trials, this error proves to be not so catastrophic for temperatures of at least 1, though we would anticipate linear regret after more time periods. We find that increasing $\kappa_{\text{sampling}} > 1$ yields exploratory behavior more aligned with TS where optimal actions more likely to be taken in the later time periods and and there is a more gradual reduction of probability mass from other actions (see Appendix B).

### 4.1.2 NATURAL LANGUAGE BANDIT

To demonstrate one concrete instance of how our proposed LLM-based PSRL may meet the demands of a real-world decision-making problem, we adapt the **customer service task** of Tajwar et al. (2025) into a multi-armed bandit problem. In each of $K = 20$ total time periods, the agent may either ask a question or offer a solution to address a customer issue randomly sampled from the dataset[3] of Tajwar et al. (2025). Similar to Tajwar et al. (2025), we use two additional LLMs to simulate the customer (who answers the agent's questions and tries suggested solutions as a non-technical person would) and to be a judge/reward function who ultimately determines the binary reward indicating successful resolution of a customer's issue. All models use GPT-4o as the underlying LLM.

For our LLM-based PSRL, we consider two methods for specifying the prior distribution that PSRL takes as input. In the first case, we simply ask GPT-4o to provide a prior distribution (a list of plausible underlying issues for the customer complaint as well as guessed probabilities based on how likely the model perceives the issue to be) that is given directly as input to our LLM-based PSRL agent. In preliminary experiments we found that, while this agent is capable of finding success often, it can suffer from issues of prior misspecification, where the true solution (also given in the dataset of Tajwar et al. (2025)) is not within the support of the LLM-generated input prior. To remedy this without giving away the answer, we use a second method of generating an input prior that guarantees it is well-specified; we provide the dataset solution for the sampled customer service issue to GPT-4o and indicate that it is one possible resolution but that GPT-4o must itself assign a probability to it based on how plausible it is perceived to be. We report the results of this latter agent as "well-specified" in Figure 5, where all agents were run for a total of 20 trials.

In the face of prior misspecification — something that the base PSRL algorithm does not entertain by assumption and, therefore, has no explicit mechanism to cope with — baseline LLM agent designs still cannot achieve a statistically-significant improvement over PSRL. Furthermore, once the prior misspecification is removed (without handing the solution away as the agent must still sift through other plausible sources of customer issues), PSRL is able to demonstrate strong exploration that far exceeds baseline methods on a real-world task with a tremendously-large action space.

### 4.2 TABULAR MDPS

For a tabular MDP widely known as a hard exploration task, we turn our focus to a truncated variant of the **RiverSwim** environment (Strehl & Littman, 2008). RiverSwim is a six-state chain where the agent begins in the leftmost state. The stochastic transition function mimics a water current that allows an agent to deterministically swim to the left (downstream with the current) but only stochastically swim to the right (upstream against the current) with a 35% chance of success and a small 5% chance of being pushed back one state downstream (Osband et al., 2013). Swimming downstream in the initial state results in a small reward of 0.005. Successfully swimming all the way upstream allows the agent to reach the rightmost state where it can collect a reward of 1. As all other rewards are zero, a RiverSwim agent must explore the full length of the river to learn optimal behavior. To keep financial costs down, we truncate the environment to a river of length 3 (one initial state, intermediate state, and terminal state) with $H = 6$.

We compare our LLM-based implementation of PSRL with a vanilla PSRL agent for a tabular MDP (Osband et al., 2013). The latter models epistemic uncertainty over the transition function as a collection of $|\mathcal{S}||\mathcal{A}|$ Dirichlet distributions. This epistemic state representation allows for the

---

[3] https://github.com/tajwarfahim/paprika/blob/main/llm_exploration/game/game_configs/customer_service.json

computational conveniences of Dirichlet-multinomial conjugacy. We further model unknown rewards with a discrete uniform prior over $\{0, 0.005, 1\}$. Cumulative regret curves shown in Figure 6 compare our LLM-based PSRL with a `Dirichlet(0.1,0.1,0.1)` prior against vanilla PSRL (with the standard uniform Dirichlet prior initialization of $\alpha_0 = \frac{1}{|\mathcal{S}|}$). We use $\kappa_{\pi^\star} = \kappa_{\text{posterior}} = \kappa_{\text{sampling}} = 1$ and all agents are run for 40 independent trials, except the vanilla PSRL agent run for 1,000. We also compare against the LLM agent baselines of Reflexion and ICRL with $p = 1$.

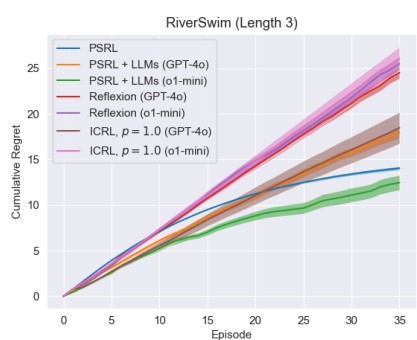

Our initial results with RiverSwim were negative (see Appendix C) as GPT-4o struggled to cope with maintaining and updating the verbose epistemic state representation describing reward information and next-state transitions across all 12 state-action pairs. Curiously, however, this negative result provided an opportunity to assess a claim of Section 3.2 that more-capable LLMs would allow our PSRL implementation to scale gracefully to more complex tasks. Indeed, by upgrading from GPT-4o to o1-mini, Figure 6 shows that our LLM-based PSRL is capable of achieving sub-linear regret on par with vanilla PSRL. Reflexion is unable to persevere past failed attempts to swim upstream before settling for the smaller downstream reward of $0.005$. ICRL has just over 25% of trials where it stumbles into the optimal policy and sticks with it while, for 60% of trials, it too falls back to pursuing the downstream reward. Moreover, the same LLM upgrade has little impact on the performance of Reflexion and actually manages to worsen the performance of ICRL; for the latter, we suspect the performance degradation stems from a combination of the stochastic transition dynamics coupled with the large quantity of ICL demonstrations that perhaps mesh poorly with the reasoning steps of o1-mini. Nevertheless, we find that LLM planning issues re-emerge in our LLM-based PSRL upon scaling up to a larger instance of RiverSwim (see Appendix D).

Figure 6: Cumulative regret curves for the RiverSwim environment with 3 states. Labels show the choice of constituent LLM model (GPT-4o or o1-mini) in each LLM agent.

## 4.3 NATURAL LANGUAGE MDPS

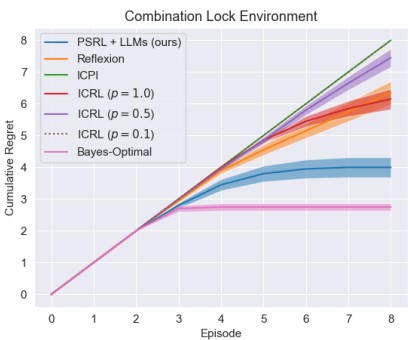

Figure 7: Cumulative regret curves for the combination lock environment. The vertical axis shows turns to identify the unlock code.

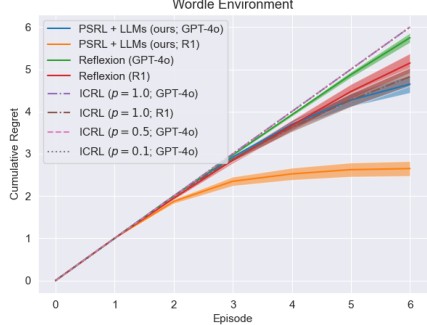

Figure 8: Cumulative regret curves for the Wordle environment. Labels show the choice of constituent LLM model (GPT-4o or DeepSeek-R1) in each LLM agent.

Having verified that our LLM-based PSRL retains efficient exploration in more traditional environments, we now turn to tasks entirely inaccessible by classic PSRL. The first of these tasks is a **combination lock** environment where an agent must enter $H = 3$ distinct digits in order to open a lock and receive a reward of $+1$. All other rewards are zero and the agent is provided with (verbal) state information indicating whether the most recently guessed digit is either in the correct position

for the correct code, present in the correct code but in some other position, or simply not present in the correct code at all. An agent has $K = 8$ episodes to identify the correct combination and, with each one of 20 independent trials having an unlock code sampled uniformly at random from all 720 possible codes, exploration via uniform random code selection has below $0.14\%$ chance of success.

The second task is the challenging web game known as **Wordle** (Lokshtanov & Subercaseaux, 2022), where an agent has exactly $K = 6$ episodes to enter $H = 5$ distinct letters (which need not be a dictionary word) that spell a correct target word and receive a reward of $+1$. Across 40 trials (except ICPI run for 10 trials due to its significantly higher financial cost and lengthy run times), the target word is chosen uniformly at random from a filtered corpus of English dictionary words. The agent is provided verbal feedback in each state indicating whether the most recently guessed letter is in the correct position for the target word, in the target word but at some other position, or not present in the target word at all.

Our LLM-based PSRL agent ($\kappa_{\text{sampling}} = \kappa_{\pi^\star} = \kappa_{\text{posterior}} = 1$) is given an uninformative prior which describes all non-repeating codes/English words with the appropriate length as being equiprobable; the unlock code/target word is an environment proxy (Lu et al., 2023) such that knowledge of the proxy is a sufficient statistic for recovering the full MDP. For the combination lock, we also compute the Bayes-optimal policy with respect to the same uninformative prior and plot its cumulative regret for comparison. To assess the efficacy of our LLM-based PSRL with another alternative choice of constituent LLM, we present Wordle results with DeepSeek-R1 (Guo et al., 2025).

The combination lock and Wordle environments represent distinct instances of an exploration problem at differing scales within a deterministic environment. Notably, the immediate per-digit/letter feedback eliminates the challenge of credit assignment entirely (as there is no ambiguity in how each decision impacts delayed rewards) and isolates exploration as the sole data efficiency obstacle. Our results (Figures 7 and 8) show that the LLM-based PSRL is able to most effectively explore the space of possible unlock codes/target words relative to the baseline methods. Crucially, none of the three constituent LLMs used by PSRL are prompted to explicitly encourage exploration. Rather, these results further illustrate how prompting these LLMs to perform atomic functions of PSRL and allowing the algorithm to prescribe how those outputs should be orchestrated in the agent design can yield an effective exploration strategy. In Wordle, we observe that DeepSeek-R1 provides a performance improvement to all LLM agents; however, we find that its enhanced reasoning capabilities applied to even our best baseline LLM agent are insufficient to yield a statistically-significant improvement over our LLM-based PSRL, even when run with a less-capable GPT-4o as the constituent LLM. We invite readers to see Appendix E for analogous results on combination lock with DeepSeek-R1.

The ICPI paper (Brooks et al., 2023) includes a dataset balancing scheme for ICL, presuming the requisite data has already been collected. While reasonable for some environments, exploration is fundamentally about governing data collection to synthesize optimal behavior and, in these domains, ICPI never observes non-zero reward and collapses to a random policy. For ICRL, using all available data with $p = 1$ is equivalent to the "LLM policy" evaluated by Klissarov et al. (2025), who also find poor performance in Wordle. While results in the combination lock domain are better, we find that decreasing the keep probability $p$ is detrimental to the "exploratory" ICRL of Monea et al. (2024). In Reflexion, we observe that self-reflections during the early stages of learning generically encourage exploration of untested digits/letters, assuming the agent knows how to explore upon simply being instructed to do so. Only once uncertainty has largely been resolved do reflections become specific suggestions about how to explore with particular digits/letters and their ordering.

## 5 CONCLUSION

While much of the burgeoning literature surrounding LLM agents has felt compelled to design new algorithms for solving RL problems, we here have demonstrated that an existing algorithm, PSRL, can be implemented with LLMs. The main advantage of our proposed LLM-based implementation of PSRL is allowing agent designers to leverage the strong generalization and reasoning capabilities of LLMs in natural-language environments while simultaneously capitalizing on the well-studied exploration properties of TS. Future work might extend regularization methods (Jiang et al., 2015; Arumugam et al., 2018; Rathnam et al., 2023) that embrace inaccurate transition models to rectify deficiencies we observed with LLM planning in stochastic domains. Our preliminary results (see Appendix F) on recovering information-directed exploration (Russo & Van Roy, 2018) with LLMs

represent what is likely to be another very fruitful direction for future work and further reinforces the potential benefits of implementing, rather than replacing, existing RL algorithms with LLMs.

## ETHICS STATEMENT

The impact of LLMs in recent years has been undeniable and so immense as to extend beyond the confines of the machine learning community, drawing scrutiny from the broader public. As this paper studies mechanisms for improving the decision-making capabilities of LLMs that are becoming increasingly more capable and ubiquitously deployed, there is potential for broad impact stemming from our work. This impact is amplified by the fact that our contributions for improved exploration in LLMs center around Thompson sampling (Thompson, 1933), an exploration strategy whose impact in real-world decision-making problems such as recommendation systems (Chapelle & Li, 2011) and beyond (Russo et al., 2018) is already well known.

## REPRODUCIBILITY STATEMENT

For all LLM agents evaluated in our experiments, the key items needed to reproduce our results are the system prompts, user prompts, environment descriptions, environment details, and the process by which constituent LLMs are queried and have their outputs organized. All of these details can be found across Section 4 and Appendix H along with a rough (anecdotal) estimates of the associated financial cost of running these experiments in Appendix I. Details of all evaluation domains can be found in Section 4 and the associated natural language descriptions common to all LLM agents evaluated in this work can be found in the appropriate sub-sections of Appendix H. As this paper relies heavily on API access to LLMs, it is impossible to obtain granular details on how much compute was used by our experiments. Instead, we have included Appendix I with ballpark estimates of how many tokens were used by our proposed approach in each of our evaluation domains as well as a translation of those token counts to dollar costs.

### ACKNOWLEDGMENTS

This work was supported by ONR MURI N00014-24-1-2748, ONR grant N00014-23-1-2510, and Azure credits from a Microsoft AFMR grant. We gratefully acknowledge Ilia Sucholutsky for setup and debugging assistance in our experiments. We thank Ted Sumers for a helpful suggestion to use XML formatting when processing trajectories with LLMs. Finally, we thank Ilia Sucholutsky and David Abel for feedback and insightful comments on an early draft of the paper.

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

# A    RELATED WORK

While our primary focus in this paper is on efficient exploration for LLM agents, the broader challenge of efficient exploration for RL agents is a long-studied topic. One route to achieving statistically-efficient exploration relies on the use of "optimism in the face of uncertainty," where approaches either implicitly or explicitly maintain over-inflated value function estimates for all state-action pairs (Kearns & Singh, 2002; Brafman & Tennenholtz, 2002; Kakade, 2003; Auer et al., 2009; Strehl et al., 2009; Jaksch et al., 2010; Dann & Brunskill, 2015; Azar et al., 2017; Dann et al., 2017; Jin et al., 2018; Zanette & Brunskill, 2019; Dong et al., 2022). These optimistic biases are calibrated by an agent designer to incentivize agent visitation of each state-action pair sufficiently many times and eventually result in accurate value estimates that give rise to optimal behavior. Nie et al. (2024) attempt to realize such an optimistic exploration strategy with LLMs (specifically, combining UCB (Auer et al., 2002) with Gemini (Team et al., 2023)) for multi-armed bandit problems and demonstrate the difficulty in coupling statistical machinery like confidence intervals with LLMs outright. While our proposed implementation relies on an equally (if not more) complex statistical object, the Bayesian posterior, our experiments suggest that LLMs in certain cases may maintain an approximation sufficient for guiding exploration.

Existing designs for LLM agents either do not explicitly engage with the challenge of exploration or do so with complete reliance on in-context learning (ICL) (Brown et al., 2020). One of the most popular LLM agent designs is Reflexion (Shinn et al., 2024) where the policy LLM charged with selecting actions is informed at each episode by a "self-reflection" generated from another LLM given the previous episode trajectory. While suitable for some tasks, we observe in our experiments that the self-reflection LLM often "passes the buck" and encourages exploration generically in language without providing a clear strategy for the downstream policy LLM to do so. By relying on LLMs to provide the requisite functions for implementing a prudent choice of existing RL algorithm, we encounter strategic exploration without needing to explicitly instruct any of the involved LLMs to explore.

LLM agents that rely on ICL to enable exploration follow suit with a line of work that examines Transformer-based RL agents in non-natural-language tasks (Laskin et al., 2022; Liu et al., 2023; Lee et al., 2024b; Dai et al., 2024; Yan et al., 2025). These methods often rely on casting ICL as either implicit, approximate Bayesian inference (Xie et al., 2022; Zhang et al., 2023) or within the "control as inference" framework (Levine, 2018); one key challenge with the former is that such implicit posterior knowledge cannot be flexibly and explicitly leveraged to guide exploration, whereas the latter suffers from not capturing epistemic uncertainty at all (O'Donoghue et al., 2020; Tarbouriech et al., 2023). Very close to the spirit of our work is the in-context policy iteration (ICPI) method of Brooks et al. (2023), who take the classic RL algorithm of policy iteration (PI) (Howard, 1960) and implement it with LLMs and ICL. Unfortunately, the original PI algorithm is oriented towards tabular MDPs that allow for iterating over all state-action pairs simultaneously. While the ICPI algorithm forgoes this in favor of online data collection and resampling via experience replay (Lin, 1992), the authors find it necessary to sample with a dataset balancing scheme to ensure the accuracy of ICL; this presumes that the "right" data is already present or easily acquired from the environment. In larger environments where data must be judiciously acquired, we find that ICPI is never able to collect the data needed for ICL to exhibit any kind of performant behavior. Monea et al. (2024) study a selective "dropout" strategy for the ICL demonstrations used by a policy LLM. However, such a strategy mirrors $\epsilon$-greedy exploration (Watkins & Dayan, 1992) without making a concerted effort to strategically guide decision-making, much like how classic dropout in deep RL (Gal & Ghahramani, 2016) is a poor proxy for uncertainty-based exploration (Osband, 2016b). In contrast to ICL, the core idea studied in this work is conceptually similar to meta-prompting (Goodman, 2023), where an agent incrementally accumulates salient environmental knowledge within its system prompt to refine behavior in each episode; while prior work has suggested that meta-prompting is an implicit approximation of posterior sampling (Fränken et al., 2023), we here are exclusively concerned with the explicit implementation of PSRL.

A related line of approaches examines using classic (deep) RL methods in tandem with LLM reward functions (Klissarov et al., 2025; Kwon et al., 2023; Zheng et al., 2024). These approaches, while interesting, largely focus on non-linguistic domains whereas our goal is to bring ideas on data-efficient RL to bear on the natural language domains where LLMs stand to have the most impact.

The posterior-sampling-based exploration strategy we consider in this work connects more broadly to initial investigations surrounding the information gathering capabilities of LLMs (Ke et al., 2024).

Lastly, we note that the Reinforcement Learning from Human Feedback (RLHF) pipeline (Stiennon et al., 2020; Ouyang et al., 2022) used to explicitly optimize LLMs also faces an underlying sequential decision-making problem (in the original formulation, a contextual dueling bandit (Yue et al., 2012; Dudík et al., 2015)) and, as such, may greatly benefit from mechanisms to facilitate efficient exploration (Xu et al., 2023; Dwaracherla et al., 2024). Concretely, at any point in the fine-tuning process either by RLHF or Reinforcement Learning from AI Feedback (RLAIF) (Lee et al., 2024a), there will be preference data that offer very little utility or change in LLM responses and those that stand to dramatically improve response quality. By actively exploring for the latter kind of prompts and responses, one stands to arrive at a more proficient LLM with fewer iterations of RLHF or RLAIF. While such work is nascent, our results may offer a promising new pathway for LLMs to achieve the strategic exploration that could reduce these significant data burdens.

## B    MULTI-ARMED BANDIT RESULTS

### B.1    BERNOULLI BANDIT

As noted by Krishnamurthy et al. (2024), the financial and temporal costs of running LLM agents can be quite significant. With only 20 trials, it would be presumptuous to make any sweeping claims about superior performance of one method relative to others. Fortunately, the goal of our multi-armed bandit experiment is aimed at at a relativistic comparison in the quality of exploration with our LLM-based PSRL relative to classic TS. To this end, we borrow the surrogate statistics employed by Krishnamurthy et al. (2024) to provide deeper insight into the long-term exploratory behavior of LLM-based PSRL. Figure 9 reports the *suffix failure* frequency, where a suffix failure at time period $t$ is a binary statistic defined as 1 if the optimal action $A^\star$ is never chosen in time periods $[t, T]$ and 0 otherwise. Clearly, an agent experiencing a large number of suffix failures early on in learning would be unlikely to identify $A^\star$ when run for a larger number of time periods. Figure 10 reports the (scaled) *minimum action frequency*, which reports at time period $t$ the frequency of the least-chosen action in the first $t$ time periods: $\frac{1}{t} \cdot \min_{a \in \mathcal{A}} \left| \{ A_{t'} \mid t' \in [t], A_{t'} = a \} \right|$. The statistic is scaled by $|\mathcal{A}|$ to reside in $[0, 1]$. As an agent's knowledge of the world accumulates, one would naturally expect an agent to gradually cease selection of some (ideally, sub-optimal) actions and incur lower minimum action frequencies. Together, these two surrogate statistics paint a picture of whether or not the exploration of a LLM bandit agent gravitates toward $A^\star$ over time.

Notably, we find that increasing the temperature $\kappa_{\text{sampling}}$ of the posterior sampling LLM has profound impact on how well our LLM-based PSRL explores according to these metrics. In particular, we find that increasing $\kappa_{\text{sampling}}$ leads to exploratory behavior more closely aligned with that of classic TS compared to lower temperatures values.

### B.2    CUSTOMER SERVICE BANDIT & PRIOR (MIS)SPECIFICATION

To demonstrate one concrete instance of how our proposed LLM-based PSRL might meet the demands of a real-world decision-making problem, we adapt the customer service task of Tajwar et al. (2025) into a multi-armed bandit problem. In each of $K = 20$ total time periods, the agent may either ask a question or offer a solution to address a customer issue randomly sampled from the dataset[4] of Tajwar et al. (2025). Similar to Tajwar et al. (2025), we use two additional LLMs to simulate the customer (who answers the agent's questions and tries suggested solutions as a non-technical person would) and to be a judge/reward function who ultimately determines the binary reward indicating successful resolution of a customer's issue. All models use GPT-4o as the underlying LLM.

For our LLM-based PSRL, we consider two methods for specifying the prior distribution that PSRL takes as input. In the first case, we simply ask GPT-4o to provide a prior distribution (a list of plausible underlying issues for the customer complaint as well as guessed probabilities based on how likely the model perceives the issue to be) that is given directly as input to our LLM-based

---

[4]https://github.com/tajwarfahim/paprika/blob/main/llm_exploration/game/game_configs/customer_service.json

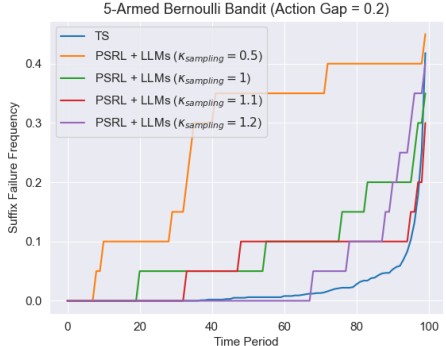 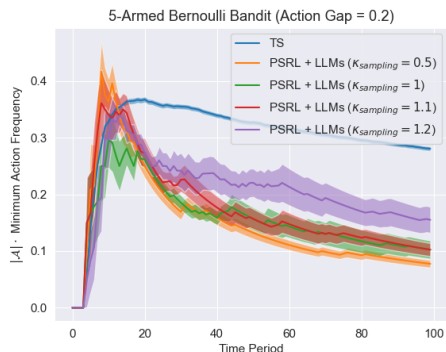

Figure 9: Suffix failure frequency for a 5-armed Bernoulli bandit with $\Delta = 0.2$. A suffix failure occurs at time $t$ if $A^\star$ is never chosen in time periods $[t, T]$.

Figure 10: Scaled minimum action frequency for a 5-armed Bernoulli bandit with $\Delta = 0.2$. At time period $t$, this is the average frequency of the least-chosen action in time periods $[1, t]$.

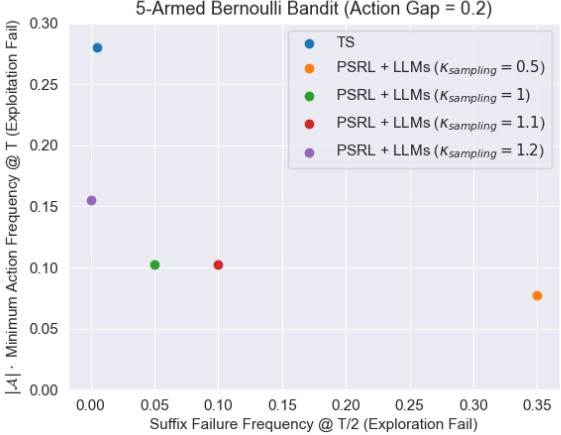

Figure 11: A scatter plot of suffix failure frequency vs. minimum action frequency for Thompson sampling and our LLM-based PSRL with varying $\kappa_{\text{sampling}}$.

PSRL agent. In preliminary experiments we found that, while this agent is capable of finding success often, it can suffer from issues of prior misspecification, where the true solution (also given in the dataset of Tajwar et al. (2025)) is not within the support of the LLM-generated input prior. To remedy this without giving away the solution, we use a second method of generating an input prior that guarantees it is well-specified; we provide the dataset solution for the sampled customer service issue to GPT-4o and indicate that it is one possible resolution but that GPT-4o must itself assign a probability to it based on how plausible it is perceived to be. We report the results of this latter agent as "well-specified" in Figure 5. All agents were run for a total of 20 trials.

In the face of prior misspecification, something that the base PSRL algorithm does not entertain by assumption and therefore has no explicit mechanism to cope with, baseline LLM agent designs still cannot achieve a statistically significant improvement over PSRL. While theory is not a focus of this work, we simply note in passing that prior misspecification of posterior-sampling methods is a well-studied topic in bandit learning (Russo & Van Roy, 2014; Simchowitz et al., 2021; Liu et al., 2022), where one can provably expect a graceful degradation in performance commensurate with the degree of misspecification; colloquially, similar results are expected for the full RL setting as discussed, for instance, in the introduction of O'Donoghue (2021). Future work may greatly benefit from expanding on our results to more carefully examine how PSRL can remain robust in the face of such misspecified priors. Moreover, once the prior misspecification is removed (without handing the

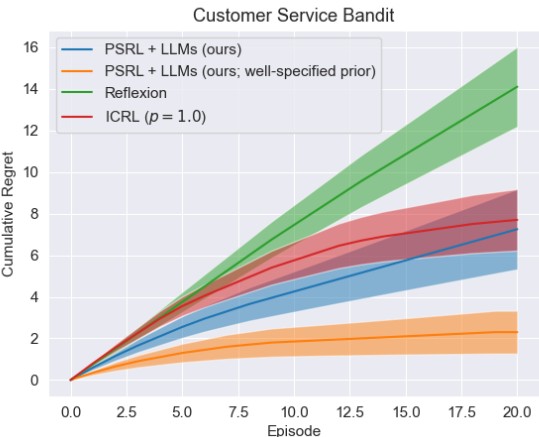

Figure 12: Cumulative regret curves for the real-world customer service bandit task. All LLM agents use GPT-4o.

solution away as the agent must still sift through other plausible sources of customer issues), PSRL is able to demonstrate strong exploration that far exceeds baseline methods on a real-world task with a tremendously large action space.

## C  EARLY FAILURES WITH GPT-4O IN RIVERSWIM

As RiverSwim is a stochastic environment, even a limited number of states may still demand a significant episode horizon in order to provide even a chance of learning progress. To keep the financial costs of our RiverSwim experiments down with horizons as small as 6 and as large as 50, we employ a policy caching scheme that capitalizes on the underlying tabular MDP that is RiverSwim. In particular, the policy LLM of *all* LLM agents (ours and baselines) used in each episode only makes one API call per *novel* state visited and the resulting selected action is cached for that state; if a state is ever revisited within the same episode, then this cached action is automatically reused without making an additional policy LLM call. After an episode is completed, this cache is then cleared and reset for the next episode. Notably, as the optimal policy for RiverSwim is non-stationary (since, if the agent is unsuccessful in swimming upstream towards the end of the episode, it is optimal to turn around and collect the smaller downstream reward), this means that the cumulative regret curves across all agents are potentially worse than what they would have been if the agents were allowed to act in a non-stationary fashion. Nevertheless, as there are only two actions in the MDP, we anticipate that the impact of this cost-saving measure on our results is minimal and equitable across all evaluated agents.

In Section 4.2, we reported positive results in a truncated (length-3) variant of the classic RiverSwim environment (Strehl & Littman, 2008) *upon switching* from GPT-4o to o1-mini as the underlying LLM for our PSRL implementation. For clarity, we use this section to detail the initial failures we encountered with GPT-4o in RiverSwim. Figure 13 shows the associated cumulative regret curves adhering to the same setup as outlined in Section 4.2, except we use $\kappa_{\pi^\star} = \kappa_{\text{posterior}} = 0$ and $\kappa_{\text{sampling}} = 0.5$. Despite achieving the best regret curve out of all presented LLM agents in RiverSwim, both of our LLM-based PSRL variants with GPT-4o incur near-linear regret while most instances of classic PSRL are able to achieve optimal behavior.

We also report both vanilla and LLM-based PSRL run with prior distributions where all deterministic RiverSwim transitions (only those where the agent swims downstream) are given as prior knowledge. We posited that supplying all deterministic transitions as prior knowledge would fare better against classic PSRL. While this does allow LLM-based PSRL to exhibit optimal behavior in many trials, far too many still fail as the optimal policy LLM struggles to select optimal actions, even when supplied with posterior samples that have high fidelity to the true environment. Reasons for this include misread transition probabilities (such as swapping numerical values of the input posterior

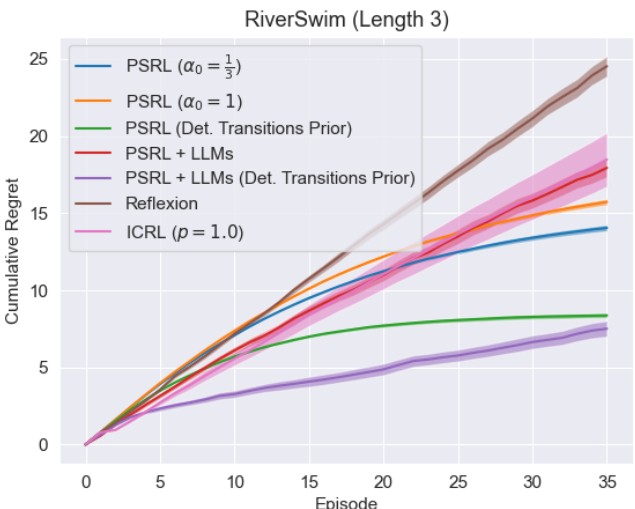

Figure 13: Cumulative regret curve for the RiverSwim environment with 3 states. Algorithms with knowledge of all deterministic transitions supplied *a priori* are labeled.

sample) as well as a lack of understanding for long-term planning. Additionally, we observe a rare occurrence where posterior updates can be prone to catastrophically forgetting a single transition, thereby halting learning progress entirely should the omitted transition be essential to reaching the upstream reward.

## D    LIMITATION: SCALING UP STOCHASTIC ENVIRONMENTS

While the success of our LLM-based PSRL in RiverSwim after upgrading to o1-mini from GPT-4o is encouraging, we find that the scalability of such a substitution is short-lived. Recall that our version of RiverSwim used in the preceding section is a truncated variant down to a length-3 river. Unfortunately, as seen in Figure 14, just increasing the river by one additional intermediate state to obtain a length-4 RiverSwim environment ($H = 20$) causes the performance of our LLM-based PSRL to degrade into linear regret.

This negative result underscores a crucial distinction in the choice of epistemic state between agents; that is, the statistical object $\text{Dirichlet}(0.1, 0.1, 0.1, 0.1)$ used by classic PSRL and the natural language string `Dirichlet(0.1,0.1,0.1,0.1)` used in LLM-based PSRL. For deterministic transitions in RiverSwim, classic PSRL is able to see eventual concentration to a Dirac delta distribution. Meanwhile the LLM-based PSRL agent, while successful at maintaining visitation counts, is slow to achieve the same convergence and, across many posterior samples, leaves non-negligible probability mass on non-existent transitions with fictitious rewards. One plausible explanation would be that such concentration errors stem from a lack of familiarity by the LLMs, given that Dirichlet distributions with fractional parameters are encountered with less frequency (McCoy et al., 2024); however, our preliminary experiments with a `Dirichlet(1,1,1,1)` prior showed no significant improvement.

Issues with posterior concentration notwithstanding, we also find that far too many episodes fail as the optimal sample policy LLM struggles to select optimal actions, even when supplied with posterior samples that have high fidelity to the true environment. Even with chain-of-thought prompting, we find a clear lack of understanding for long-term, value-based planning; the preliminary success with length-3 RiverSwim suggests that this failure is connected to the increased verbosity of the epistemic state that, in turn, compromises the optimal sample policy LLM's ability to account for the value of traversing the full river over collecting the small downstream reward repeatedly. Altogether, while the overall result is negative, we anticipate that these issues may resolve organically in a manner similar to our early challenges with GPT-4o in length-3 RiverSwim; that is, by leveraging a more advanced alternative LLM. Even if recent open-source reasoning models (Jaech et al., 2024; Guo et al., 2025)

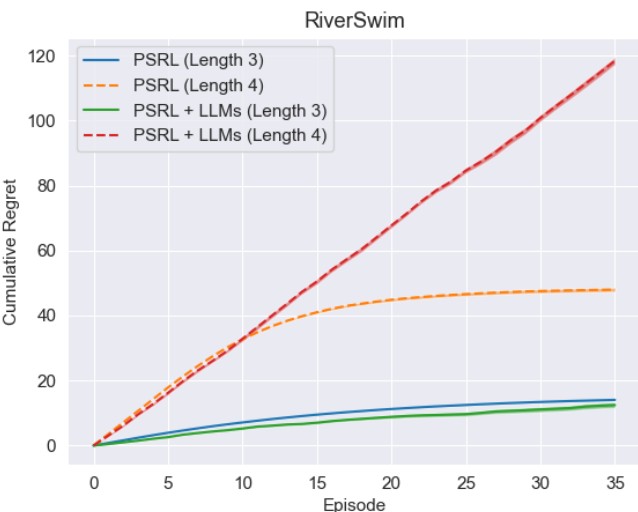

Figure 14: Cumulative regret curves for the RiverSwim environments with 3 (solid lines) and 4 (dashed lines) states, respectively. o1-mini is used exclusively with our LLM-based PSRL.

prove ineffective at fulfilling this purpose, one might still naturally anticipate that such deficiencies will disappear with time assuming future LLM capabilities continue to expand.

## E  ADDITIONAL DEEPSEEK-R1 RESULTS

While our experiments with RiverSwim (Figure 6) confirm the benefits of reasoning models that invest additional computational effort to produce so-called "reasoning" tokens prior to emitting response tokens, models such as o1-mini can be prohibitively expensive. To reduce these financial burdens and assess the efficacy of our proposed LLM-based PSRL with an alternative choice of constituent LLM, we present results for the combination lock (Figure 15 – 20 trials) and Wordle (Figure 8 – 40 trials) environments with DeepSeek-R1 (Guo et al., 2025).

Our results aggregated across both domains yield two key observations. At the highest level, we observe that R1 provides a performance improvement to all LLM agents (both ours and baselines). Curiously, we find that this performance improvement varies by model and domain; across both environments, we see very small improvements in Reflexion. Meanwhile, performance improvements for ICRL in the combination lock task and our LLM-based PSRL in Wordle are significant. More importantly, we find that the enhanced reasoning capabilities of DeepSeek-R1 applied to our best baseline LLM agents is not sufficient to yield a statistically-significant improvement over our proposed LLM-based PSRL, even when run with a "weaker" or less-capable GPT-4o as the constituent LLM. Such a result is somewhat reminiscent of classic boosting (Freund & Schapire, 1997), wherein an ensemble of weak learners are composed together into a strong (supervised) learner. Furthermore, these empirical results might (loosely) suggest that the strategic exploration strategy (specifically, Thompson Sampling) forged into the design and structure of the PSRL algorithm offers something beyond what a current strong reasoning model is capable of today, especially when given the freedom in action selections afforded by a LLM agent design like ICRL.

## F  LIMITATION: BEYOND THOMPSON SAMPLING

While PSRL, through the use of TS, is known to yield a strong exploration strategy, it is by no means perfect. In the bandit literature, shortcomings of TS are well-known and naturally become more salient in the full RL problem (Russo & Van Roy, 2018; Lu et al., 2023). By only executing actions with some probability of being optimal, TS will never take sub-optimal actions that may yield tremendous information gain. Figure 3 already illustrates how a PSRL agent's uncompromising

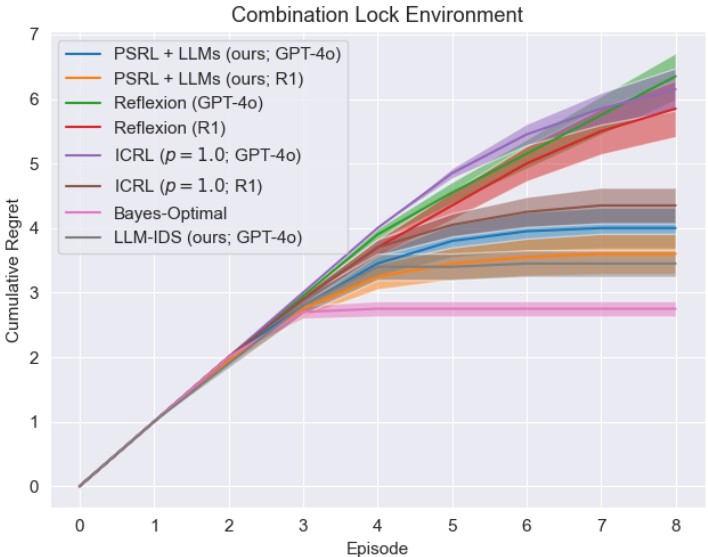

Figure 15: Cumulative regret curves for the combination lock environment. Labels show the choice of constituent LLM model (GPT-4o or DeepSeek-R1) in each LLM agent.

execution of potentially-optimal policies cripples exploration and solely allows for the testing of two unknown letters at a time.

One remedy is to seek out instantiations of information-directed sampling (IDS) (Russo & Van Roy, 2018). IDS is an algorithmic design principle that advocates for using a policy which balances between performance shortfall and information gain. While supported by a rigorous corroborating theory in both bandits and RL (Lu et al., 2023), concrete and practical instantiations of IDS are difficult to come by on account of the challenges surrounding information gain estimation (McAllester & Stratos, 2020). Moreover, the temporally-delayed consequences absent from bandits but present in RL problems pose an additional challenge as a proper IDS agent must forecast future opportunities for knowledge acquisition several steps into the future when evaluating current actions.

We present an initial design for a IDS agent with LLMs. Our proposed LLM-IDS agent is myopic in that it only takes immediate information gain about optimal behavior at the next timestep into account. Nevertheless, the feedback structure of the combination lock environment allows such an agent to be unconcerned with temporally-delayed information. For a current state $s_h \in \mathcal{S}$, we define two $|\mathcal{A}|$-dimensional vectors, $\rho$ and $\mathcal{I}$, where $\rho(a) = \mathbb{E}\left[V^\star_{\mathcal{M},h}(s_h) - Q^\star_{\mathcal{M},h}(s_h, a)\right]$ is the expected regret of taking action $a \in \mathcal{A}$ in $s_h$ under the agent's current posterior and $\mathcal{I}(a) = \mathbb{I}(\pi^\star; R_h, S_{h+1} \mid A_h = a, S_h = s_h)$ is the information gained (formally, the conditional mutual information (Cover & Thomas, 2012)) about the optimal policy by taking action $a$ from state $s_h$. IDS calls for sampling an action from the distribution that minimizes the information ratio: $\min_{\pi \in \Delta(\mathcal{A})} \frac{\mathbb{E}_{a \sim \pi}[\rho(a)]^2}{\mathbb{E}_{a \sim \pi}[\mathcal{I}(a)]}$. Normally, computation of the $\rho$ and $\mathcal{I}$ vectors would be done directly with the current posterior. Instead, we recycle the same posterior update LLM from our LLM-based PSRL but incorporate two new LLMs for the provision of $\rho$ and $\mathcal{I}$; each of these LLMs is prompted on a per-action basis to assess the expected regret or information gain, respectively, from each action in the current state. With these $2|\mathcal{A}|$ LLM-generated numerical values, the convex optimization problem of minimizing the information ratio is solved to compute the policy for action selection.

We offer two empirical evaluations to highlight the limitations of LLM-based PSRL exploration inherited from TS while also underscoring the future potential of our LLM-IDS. The first is a contrived but transparent multi-armed bandit problem given as Example 2 of Russo & Van Roy (2018). In this $(K+1)$-armed informative action bandit problem, there is a unique optimal action $A^\star \in [K]$ that yields a deterministic reward of 1 while all other arms yield a reward of 0; additionally, there is an

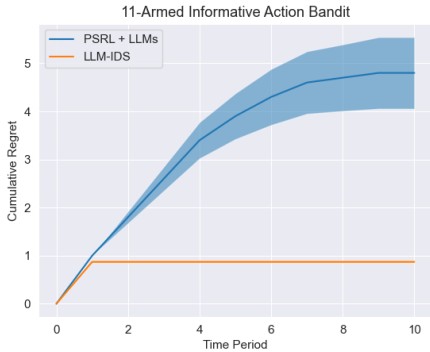 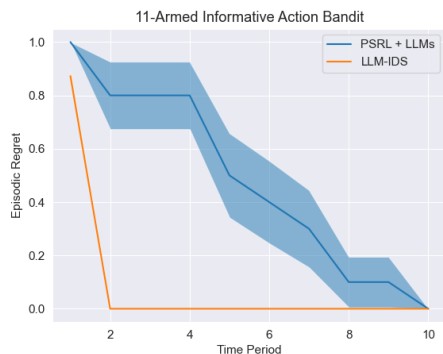

Figure 16: Cumulative regret curves for the 11-armed informative action bandit (Example 2) of Russo & Van Roy (2018).

Figure 17: Episodic regret curves for the 11-armed informative action bandit (Example 2) of Russo & Van Roy (2018).

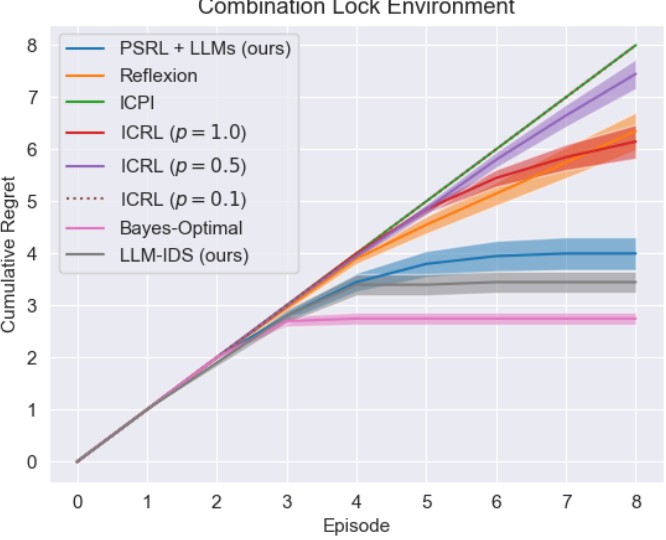

Figure 18: Cumulative regret curves for the combination lock environment including LLM-IDS.

action 0 that deterministically provides a reward equal to $(2 \cdot A^\star)^{-1}$. Naturally, an agent willing to deliberately select sub-optimal actions to gain information would take action 0 immediately and then produce optimal behavior thereafter with the identity of $A^\star$ in hand. Figures 16 and 17 show across 10 trials that LLM-IDS succeeds in recovering this optimal exploration strategy exactly for the $K = 10$ instance whereas LLM-based PSRL is incapable of doing so while exploring via TS. This result also highlights one simple instance of the flexibility that specifying natural-language priors to LLM-based PSRL affords as encoding prior knowledge about the informative action might prove difficult when limited to classic statistical distributions. Extending past this contrived yet transparent bandit example, Figure 18 shows that LLM-IDS is able to outperform LLM-based PSRL in the combination lock task by more quickly testing for unknown digits while remaining unencumbered by known digits already discovered.

## G   TOKEN EFFICIENCY

In this section, we give a brief glimpse into the token efficiency of our proposed LLM-based PSRL agent relative to our two strongest baseline LLM agents, Reflexion and ICRL ($p = 1.0$), using

GPT-4o for all constituent LLMs. Notably, our focus in this work has been exclusively on data efficiency through prudent exploration and, as such, no concerted effort has been made in either our proposed agent or baseline agents towards optimizing for token efficiency explicitly (by selecting shorter prompts as inputs to the constituent LLMs) or implicitly (by encouraging LLMs to maintain brevity in their responses). With that said, Figures 19 and 20 illustrate token efficiency of these LLM agents in the combination lock and Wordle environments by plotting cumulative regret as a function of total tokens processed (on average).

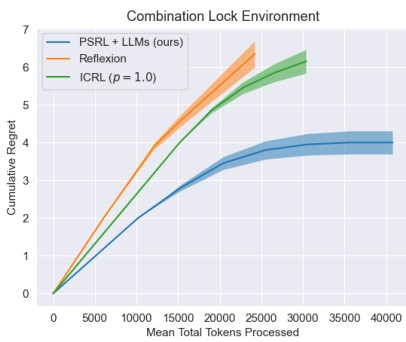

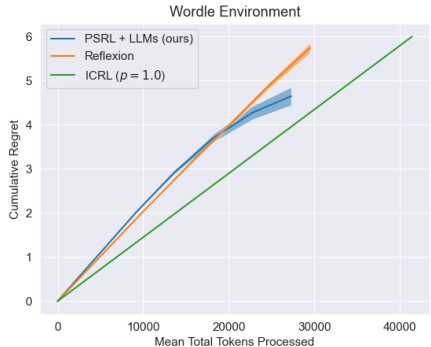

Figure 19: Cumulative regret curves for the combination lock environment as a function of total tokens processed (on average).

Figure 20: Cumulative regret curves for the Wordle environment as a function of total tokens processed (on average).

In Figure 19, we see that, despite improved performance and actual convergence towards the optimal policy, our proposed PSRL-LLM consumes more tokens (on average) than Reflexion and ICRL in the combination lock environment. We suspect the primary driver behind the excess tokens comes from the tendency of GPT-4o to fully enumerate all possible correct codes in the "posterior" — something that neither baseline agent does thereby allowing them to be more economical with respect to the total number of tokens processed. Despite that, however, we see that our LLM-based PSRL does achieve better cumulative regret even if truncated to the same number of tokens processed by either baseline agent. In Figure 20, we see that LLM-based PSRL displays token efficiency that is comparable to Reflexion and superior to ICRL in Wordle, eventually able to more consistently identify target words in fewer turns than Reflexion, resulting in lower cumulative regret. Unlike in the combination lock environment, there are far too many possibilities for possible target words and GPT-4o never even attempts to enumerate these candidates, instead opting to maintain information about candidate correct letters and positions.

## H    EXPERIMENT PROMPTS

In this section, we outline all LLM prompts used in our experiments. We will present all system prompts in orange and all user prompts in red. It is important to note that prompts are to LLM agents what typical hyperparameters (entropy regularization coefficient, PPO clip factor, batch size, *etc.*) are to deep RL agents. In that sense, prompt optimization/hyperparameter tuning of baselines is an important facet of evaluation. As is often the case when dealing with vast hyperparameter spaces, however, an exhaustive search for the best hyperparameter settings of each method evaluated would be far too onerous. Thus, while we include our prompts for all agents in our evaluation to foster reproducibility and encourage extensions of our work, we note that future work may find performance improvements with any of these LLM agents through simple refinements of these prompts for particular models and/or downstream applications.

Each LLM used in this work (both for our and baseline agents) was prompted to perform its designated function in the context of a broader agent design/algorithm (PSRL, Reflexion, ICRL, or ICPI). Thus, our prompt iteration process simply consisted of manually adjusting prompts until preliminary experiments showed the desired functionality being achieved. For baseline agents, especially those using ICL, this required few iterations; for some elements of PSRL that involve slightly more complicated entities than a policy; transition function; reward function; or evaluator,

additional iterations were needed to weed out edge cases and tack on further constraints into the initial prompt used in the first iteration. For any given domain, the ability to successfully realize the desired functionality in each of the three LLMs should serve as "unit tests" signaling to an agent designer whether or not it is sensible to run our proposed PSRL agent. More generally, we make no claim that these prompts are optimal in any sense (a claim that likely no LLM agent paper can make in good faith). Investigating these choices in prompt iteration and downstream LLM agent robustness are important areas of future research.

## H.1   LLM-BASED PSRL

In our experiments, depending on the particular environment, we consider two different forms of posterior LLM prompting. For sufficiently short horizons, the posterior LLM is given the entire trajectory in a single prompt and is expected to produce the updated posterior. For longer horizons or whenever concerns about context buffer length come into play, the posterior LLM is prompted with one full $(s, a, r, s')$ experience tuple at a time and each successive posterior becomes the prior for the subsequent update. Empirically, we find that whole trajectory updates may be more likely to result in erroneous updates where certain pieces of information may be mistakenly updated or forgotten entirely. While this becomes far less likely with per-step experience updates, the associated financial costs and time spent running the PSRL agent scale unfavorably with the horizon of the problem. We use whole trajectory observations for all LLM-based PSRL posterior updates in the RiverSwim, Combination Lock, and Wordle environments. For LLM-based PSRL multi-armed bandit results and LLM-IDS, we use per-step posterior updates.

For whole trajectory posterior updates, the approximate posterior LLM uses the following system prompt and user prompt:

> You are a Bayesian posterior distribution for a real-world sequential decision-making problem. Given a current prior belief about the environment and single trajectory observation, you should produce the posterior distribution that accurately reflects knowledge about possibly stochastic environment transitions and environment rewards based on the observed trajectory. A trajectory observation is a sequence of experiences, where each experience consists of a state, action, reward, and next state. Each unit of experience will be separated by XML <EXPERIENCE> </EXPERIENCE> tags. The posterior distribution must always be complete and describe all sources of uncertainty the agent has about the world. There can be uncertainty about a stochastic transition or reward. The posterior distribution should take into account all information provided in the observed trajectory to update the prior belief about the environment. Be direct and don't show your work. You cannot make any assumptions about the agent and the action selections used to generate the trajectory observation. Never try to model beliefs about the agent. Do not say anything beyond providing the posterior distribution. The agent's interactions with the environment will generate rewards and the posterior distribution should keep track of how any and all rewards are generated. Information and knowledge in the current prior belief about the environment should never be discarded from the posterior distribution. If there is knowledge in the current prior belief about the environment that is unaffected by the trajectory observation, then this knowledge should not be changed and must be repeated exactly in the posterior distribution. Do not say anything to distinguish between old knowledge that is being retained and updated knowledge. The environment was described to the agent like this: `<Environment Description>`

> Your current prior is as follows: `<Input prior/LLM-generated posterior>`. A trajectory observation is a sequence of experiences, where each experience consists of a state, action, reward, and next state. Each unit of experience will be separated by XML <EXPERIENCE> </EXPERIENCE> tags. Here is an observed trajectory:`<Full trajectory>`. Remember that knowledge in the current prior must only be updated but can never be discarded, forgotten, or removed. Do not say anything about which information in the posterior is new and updated or old and remains the same from the prior.

For per-step posterior updates, the approximate posterior LLM uses the following system prompt and user prompt:

> You are a Bayesian posterior distribution generator for a real-world sequential decision-making problem. A sequential decision-making problem is represented by an environment that, to each current state and action, produces a next state transition and a reward based on that transition. Transitions and rewards observed from the environment may be stochastic or may be deterministic. Given a current prior belief about the environment and single observation consisting of a next state transition and reward from the environment, you should generate the posterior distribution that accurately reflects knowledge about possibly stochastic environment transitions and environment rewards. The posterior distribution should be a complete and accurate description of all uncertainty the agent has about the world. Information from the prior belief can never be discarded, only updated to be more consistent with the given observation. The posterior distribution should take into account all information provided in the observed next state transition and reward to update the prior belief about the environment. You cannot make any assumptions about the agent and the action selections used to generate the next state transition and reward observation. Never try to model beliefs about the agent. The world may be stochastic and random such that the prior knowledge may need to be updated in the posterior distribution to be consistent with an observed transition or reward. Any knowledge in the prior belief about the environment that is not affected by the observed transition and reward should be retained in full by your posterior distribution. The environment was described to the agent like this: `<Environment Description>`

> Your current prior is as follows:`<Input prior/LLM-generated posterior>`. Here is an observed environment transition and reward:`<Single next-state transition and reward>`. Do not say anything about which information in the posterior is new and updated or old and remains the same from the prior. Whenever possible you must maintain exact, numerical probabilities.

The optimal sample policy LLM simply takes the current observation as the user prompt while using the following system prompt:

> `<Environment Description>`. Always select optimal actions that maximize value across all future states and all remaining timesteps according to the following hypothesis: `<LLM-generated posterior sample>`. You must select actions that are optimal for and perfectly consistent with the above hypothesis. For each action, you must consider its immediate expect reward as well as the expected value of future states that can be visited by selecting the action. Always select from one of the available actions to take in the environment. Just say the action after "Action: " and nothing else.

As generating a posterior sample requires specifying a full MDP, we find that the posterior sampling LLM in PSRL benefits from having distinct prompts that cater to salient aspects of generating an instance of each environment. We organize the associated environment descriptions as well as posterior sampling system prompts and user prompts by task in the following sub-sections. We also include a sub-section for all prompts used by LLM-IDS.

## H.2 MULTI-ARMED BANDITS

### H.2.1 BERNOULLI BANDIT

The environment description for the Bernoulli bandit task was given as:

You are an agent interacting with a 5-armed Bernoulli bandit problem. You have exactly 5 actions available labeled as `<List of randomly generated letters>` and each action has an independent Bernoulli distribution. When you select an action, you will receive a binary reward sampled from the associated Bernoulli distribution.

The posterior sampling LLM system prompts and user prompts were:

You are a generator of Bernoulli bandit problems. A Bernoulli bandit problem is a collection of mean reward values, one for each available action. Knowledge about the reward of each available action will be given to you in the form of a Beta distribution representing beliefs about the mean reward of each arm. This knowledge will constrain the Bernoulli bandit problems you are allowed to generate. For each action, return one plausible hypothesis for the mean reward an agent will observe when taking that action. Each mean reward you return should be consistent with the knowledge you are given about the observed rewards of each action. Each action is independent and so each hypothesis you return for the mean reward of each action will be independent of all others. You must return real, numerical values starting with the phrase "You think " and do not say anything beyond providing the mean rewards of each action. You cannot just return the mean value of the Beta distribution as your guess for the mean reward. You must return a sample from each Beta distribution as your hypothesis. Before you return your mean reward values, describe how each one obeys all constraints and knowledge provided to you. The environment was described to the agent like this: `<Environment Description>`

Your current knowledge about the mean reward of each action is as follows:`<Input prior/LLM-generated posterior>`. You must carefully read through this information to generate a Bernoulli bandit problem consistent with this knowledge.

## H.2.2 CUSTOMER SERVICE BANDIT

The environment description for the customer service bandit was given as:

You are going to role-play as a customer service agent and you have to help a customer resolve their issue. Your goal is to gather enough information to diagnose the problem and provide a correct solution. Your instructions are the following: 1.You may either ask the customer questions or suggest particular actions to the customer. 2. The customer may not be technically inclined, so keep your language simple and clear. 3.Avoid making assumptions — ask specific questions to determine the potential causes. You should guide the customer through basic troubleshooting steps and gather data on the situation. 4. You should try to make the customer satisfied and resolve their problem as quickly as possible. You should also keep your responses short and concise. 5. If the customer mentions a specific product they are using (for example, ABC electronics), then you are the customer support agent for that product/company, i.e., you represent that product or company and have to take appropriate actions without referring the customer to somewhere else. You will receive a reward of 1 if you succeed in resolving the customer's issue and all other rewards are 0. The specific scenario the customer faces is this:`<Troubleshooting task sampled from dataset>`.

The posterior sampling LLM system prompts and user prompts were:

You are a troubleshooting hypothesis generator for a customer service agent. The initial issue faced by the customer was described to the customer service agent as follows: `<Environment Description>`. The customer service agent is trying to generate hypotheses for what the customer's underlying issue really is. Given all knowledge the agent currently has currently obtained thus far about the customer's issue, you must generate a single plausible hypothesis for what the customer issue is so the agent can correctly provide the solution to the customer. Current knowledge about the customer's underlying issue will be given as a probability distribution listing possible underlying issues and the probability of those issues being accurate for the customer. The hypothesis you generate must be a sample from this distribution. While it is perfectly fine to return a sample that represents the element of the distribution with highest probability, you cannot just return the most likely hypothesis from this distribution simply because it has the highest probability. You must actually sample the distribution to generate your hypothesis. If the probabilities do not sum to 1 to form a valid probability distribution, sample a hypothesis based on what seems plausible using the knowledge available. Be as specific as possible when describing your hypothesis for the customer's issue. You cannot just vaguely state that the customer's item or some component of their item has an issue. You must be more precise than that. When you return your sample for the customer's issue, start with the phrase "You think " and do not say anything beyond providing your hypothesis.

All current knowledge about the customer's issue is as follows: `<Input prior/LLM-generated posterior>`. You must carefully read through this knowledge. Never say anything about the agent or tell the agent what to do.

Additionally, the respective system prompts used for the environment, judge/reward function, and prior-generator LLMs are as follows:

You are going to role-play as a customer experiencing a specific issue and interacting with a customer service agent. You must adhere to the following guidelines: 1. Only respond to the agent's questions and provide relevant information when prompted. 2.Your responses should be concise and reflect a typical, non-technical customer's knowledge level. 3. You are going to know what the solution for the specific issue is, but do not give away this information at any point. Your goal is to observe if the customer service agent can ask relevant questions and arrive at the specific issue. 4.When the agent asks you about a potential solution, you should determine what the outcome would be immediately (no asking the agent to wait or hold on) based on your knowledge about the true underlying problem, and let the agent know the result. For example, if your car is not starting because of lack of fuel, and the agent suggests you to check the fuel level in your car, it should fix your issue. Use your reasoning ability to decide whether the suggested solution works, given that the solution to your problem is: `<Solution to sampled dataset issue>`. 5. Reply with 'Goal reached' (and nothing more) if the agent has reached or solved your particular issue. If your issue is solved, you must reply with 'Goal reached'. Your role-play starts now. The particular problem you, as the customer, are facing is: `<Customer service issue sampled from dataset>`, and the solution to this problem is: `<Solution to sampled dataset issue>`. Note that you should never give away or tell the agent what will solve your particular problem, your goal is to see if the agent can come up with the potential solution themselves by asking relevant questions and obtaining accurate information from you.

You are the judge of a customer service agent trying to help a customer in a particular scenario. You will be given the agent's response, and you have to verify whether the agent has been able to successfully solve the problem of the customer correctly. You must adhere to the following guidelines: 1. If the agent provides a solution that is well aligned with the true solution, you must reply with <VALID>. 2. If the agent's proposed solution does not align well with the true solution, then you must reply with <NOTVALID>. 3. The agent doesn't need to match the solution word for word to be considered correct. The agent should be considered correct as long as their solution or question clearly demonstrates that the agent has correctly discovered the source of the customer's issue. 4. Prior to returning your judgement of <VALID> or <NOTVALID> think about the agent solution and true solution and provide a brief justification of why they do or do not align well. The particular scenario the customer is facing is: `<Customer service issue sampled from dataset>`, and the true solution to their problem is: `<Solution to sampled dataset issue>`.

You are the generator of a prior distribution for a Bayesian decision-making agent. The agent is faced with a customer service task described as follows: `<Customer service issue sampled from dataset>`. The agent will be given an broad initial prior as follows:
You know that rewards are binary and you will only receive a reward of 1 once the customer's issue has been resolved. If you knew all the relevant details about the source of the customer's issue, there would be no uncertainty about what correct solution to offer and obtain a reward of 1. You think that all common, reasonable issues based on the observations the customer has given are plausible. You think that more common and more realistic issues are more likely than uncommon and less realistic issues.
Your job is to provide an additional supplement to this prior that is specific to the issue the customer is facing. Give a probability distribution for the possible underlying issue a customer could be faced along with the probabilities or relative likelihood for each issue you list based on which of them are more or less likely to be the culprit. `(The next line is included if the prior is designed to be well-specified.)` Be aware that one possible issue could be `<Solution to sampled dataset issue>` and include it in your prior with a probability the appropriately reflects how plausible it is to be the issue.

### H.2.3    INFORMATIVE ACTION BANDIT

The environment description for the informative action bandit was given as:

You are an agent interacting with a `<Number of actions>`-armed bandit problem. You have exactly `<Number of actions>` actions available labeled by number as `<List of action IDs>`. When you select an action, you will receive a deterministic reward associated with that selected action.

The posterior sampling LLM system prompts and user prompts were:

You are a generator of a special class of bandit problems. A bandit problem in this class only has a deterministic reward associated with each arm. There is exactly one optimal action which yields a reward of 1. For whichever index the optimal action has, the action with index 0 must produce a reward equal to 1 divided by 2 times the optimal action index or, in other words, the reciprocal of twice the optimal action index. All other actions must produce a reward equal to 0. Knowledge about the optimal action will constrain the instance of this special bandit class that you are allowed to generate. Based on the knowledge of which actions cannot be optimal, choose one of the remaining actions to be optimal uniformly at random. Then, assign deterministic rewards to all of the actions so the bandit problem you generate belongs to the special class exactly as described. You must return real, numerical values for the deterministic action of each action starting with the phrase "You think " and do not say anything beyond providing the reward of each action. Before you return all the special bandit problem reward values, describe how each one obeys all constraints and knowledge provided to you. The environment was described to the agent like this: `<Environment Description>`

Your current knowledge about the rewards is as follows:`<Input prior/LLM-generated posterior>`. You must carefully read through this information to generate a bandit problem consistent with this knowledge that must belong to the described special class.

## H.3 RIVERSWIM

The environment description for RiverSwim was given as:

You are an agent swimming in a network of three underwater caves connected by tunnels. Each cave is labeled by its number and always has two tunnels labeled A and B that you can try to swim through. Swimming through tunnels allows you to stochastically move between the caves. There is a strong current in the water which can affect how difficult it is to successfully swim through certain tunnels. Some tunnels may be easier to swim through than others. Successfully swimming through a tunnel once in any cave does not guarantee that it will always be successful. Conversely, failing to swim through a tunnel once does not mean it is impossible and you may have to try again a few times before successfully making it through and swimming into a different cave. Swimming through specific tunnels from certain caves to reach other caves may yield scalar rewards between zero and one.

The posterior sampling LLM system prompts and user prompts were:

You are a map generator for an agent navigating an environment. The environment was described to the agent as follows:`<Environment Description>`. A map must specify exactly two pieces of information for each possible combination of current cave, tunnel, and next cave. The first piece of information is a transition probability that represents the probability of being in a specific cave, swimming through a particular tunnel, and ending up in a specific next cave. Knowledge about next cave transitions will be provided to you as a collection of Dirichlet distributions. Sampling these distributions will allow you to generate next cave transition probabilities for each cave and tunnel combination. The second piece of information is a deterministic reward that an agent will receive when being in a specific cave, swimming through a particular tunnel, and ending up in a specific next cave. You will be given knowledge about known rewards and rewards that are still unknown and uncertain. If a reward is known, you must repeat its numerical value exactly in the map you generate. If a reward is unknown, knowledge about what it could be will be given to you as a discrete uniform distribution over possible values. You will sample this distribution for each cave, tunnel, and next cave combination and include the concrete, numerical reward value in the map you generate. The input knowledge will constrain the maps you are allowed to generate and the map you generate must be consistent with the input knowledge. Any input knowledge that is known with certainty must be repeated exactly in the map you generate without modification. All transition probabilities and all rewards must be concrete, numerical values. You must sample the distributions you are given and cannot just return the mean value of any input distribution for transition probabilities or rewards. Generate the map using complete sentences starting with the phrase "You think " and do not say anything else. Do not say anything about the input knowledge from the agent including the Dirichlet and uniform distributions.

Current knowledge about the next cave transitions and rewards is as follows: `<Input prior/LLM-generated posterior>`. You must carefully read through this knowledge. Never say anything about the agent or tell the agent what to do.

## H.4 COMBINATION LOCK

The environment description for CombinationLock was given as:

You are a helpful assistant trying to guess the correct code to a combination lock as quickly as possible. The combination lock requires a three-digit code. You will incrementally construct your guess for the code that unlocks the lock by selecting one digit between 0 and 9 at each timestep. The correct code that opens the lock contains no repeated numbers. For each digit you guess, you will be given feedback indicating if the guessed digit is either in the correct position for the unlocking code, in the wrong position for the unlocking code, or does not appear in the combination lock code at all. You will receive a final reward of one if your guessed code correctly unlocks the combination lock. Otherwise, rewards will always be zero. Your only available actions are the digits from 0 to 9.

The posterior sampling LLM system prompts and user prompts were:

You are a helpful assistant trying to aid an agent in guessing an unknown code that will unlock a lock. Given all knowledge the agent currently has about the correct code, you must generate a single guess at what the correct code could be. You must read through the input information provided by the agent very carefully to produce a good, accurate guess for the correct code. The agent's current knowledge about the correct code establishes specific constraints on what your guess can be. You must generate a guess for the correct code that is consistent with these constraints. Before you return your guess, provide a short justification for each individual digit of your guess that describes how the digit is consistent with the input knowledge from the agent. When you return your guess, start with the phrase "You think " and do not say anything beyond providing your guess for the correct code. The environment was described to the agent like this: `<Environment Description>`

The agent's current knowledge about the correct code is the following:`<Input prior/LLM-generated posterior>`. You must carefully read through all information the agent has provided. Never say anything about the agent or tell the agent what decisions to make.

## H.5 WORDLE

The environment description for Wordle was given as:

You are an agent playing a customized version of the game Wordle. There is a five-letter target word from the English dictionary which you must try to guess as quickly as possible. The target word does not contain any repeated letters. You will incrementally construct your guess for this target word by selecting one letter of the alphabet at each timestep. For each letter you guess, you will be given feedback indicating if the guessed letter is either in the correct position for the target word, in the wrong position for the target word, or does not appear in the target word at all. You will receive a reward of one if your guessed word correctly matches the target word. Otherwise, rewards will always be zero. Your only available actions are letters of the alphabet.

The posterior sampling LLM system prompts and user prompts were:

You are a helpful assistant trying to aid an agent in guessing an unknown target word without any repeated letters from the English dictionary. Given all knowledge the agent currently has about the target word, you must generate a single guess at what the target word could be. You must read through the input information provided by the agent very carefully to produce a realistic, plausible guess for the target word. The agent's current knowledge about the target word establishes specific constraints on what your guess can be. You must generate a guess without repeated letters from the English dictionary for the target word that is consistent with these constraints. Before you return your guess, describe how it obeys all constraints and knowledge provided by the agent. When you return your guess from the English dictionary, start with the phrase "You think " and do not say anything beyond providing your guess for the target word. The environment was described to the agent like this: `<Environment Description>`

The agent's current knowledge about the target word is the following:`<Input prior/LLM-generated posterior>`. You must carefully read through all information the agent has provided. Never say anything about the agent or tell the agent what decisions to make.

## H.6 LLM-IDS

### H.6.1 BANDIT VERSION

As the bandit setting does not require handling of temporally delayed consequences or the provision of a current state, it is appropriate to have a separate prompting scheme for LLM-IDS.

The expected regret LLM used the following system prompt and user prompt:

> You are a pessimistic expected regret estimator for helping an agent interacting with a multi-armed bandit environment. The bandit environment was described to the agent as follows: `<Environment Description>`. You will be give the agent's current posterior distribution over the world and will also be given a candidate action. With these two inputs, you must provide a pessimistic estimate of the expected regret an agent will incur by taking the proposed action in the bandit environment. Recall that the regret of an action is the difference in the value or expected reward of the optimal policy and the value of the policy that takes the given action. The expected regret is computed by taking an expectation over the regret using the agent's current posterior distribution. Remember that the optimal policy always selects the optimal action with probability one and so you know that the value of the optimal policy is equal to 1. You must take an expectation with respect to the agent's current posterior distribution to compute expected regret. Your estimate of the expected regret incurred by taking this action in the environment must be pessimistic, which means that it is okay if the estimate you return is larger than the true expected regret but it absolutely cannot be smaller than the true expected regret. Naturally, you are being the most helpful when the expected regret estimate you provide is as close to the true expected regret as possible without going below it. You must produce a real and concrete numerical value as your estimate and say it as a decimal (no fractions) after "Final expected regret: ". Whenever possible, show calculations with concrete numbers before you give your estimate to justify it. Say nothing after "Final expected regret: " other than your estimate.

> The agent's posterior distribution reflecting knowledge and uncertainty about the world is as follows: `<Input prior/LLM-generated posterior>`. Please produce a pessimistic expected regret estimate for the following candidate action: `<Candidate action>`. If needed, round your answer to no more than three decimal places.

The information gain LLM used the following system prompt and user prompt:

> You are a conservative information gain estimator for helping an agent interacting with a multi-armed bandit environment. The bandit environment was described to the agent as follows: `<Environment Description>`. You will be given the agent's current posterior distribution over the world and will also be given a candidate action. With these two inputs, you must provide a conservative estimate of how much information the agent will gain about the optimal action of the bandit environment by taking the proposed action. Remember that information gain is computed as mutual information or the reduction between prior and posterior entropy, which is measured in bits. Your estimate of the information gained about the optimal action by taking the input candidate action in the bandit environment must be conservative, which means that it is okay if the estimate you return is smaller than the true information gain but it absolutely cannot be larger than the true information gain. Naturally, you are being the most helpful when the information gain estimate you provide is as close to the true information gain about the optimal action as possible without going over it. You must produce a real and concrete numerical value as your estimate and say it as a decimal (no fractions) after "Final information gain: ".Whenever possible, show brief calculations with concrete numbers before you give your estimate to quickly justify it. Say nothing after "Final information gain: " other than your estimate.

> The agent's posterior distribution reflecting knowledge and uncertainty about the world is as follows: `<Input prior/LLM-generated posterior>`. Please produce a conservative information gain estimate (measured in bits) for the following candidate action: `<Candidate action>`. If needed, round your answer to no more than three decimal places. Remember that sub-optimal or incorrect actions can be informative and information can be gained about the optimal action without actually selecting the optimal action. Also remember that, once the optimal action is known under the agent's posterior distribution, information gain must be equal to 0 for all actions.

### H.6.2 MDP Version

As previously mentioned, LLM-IDS retains the approximation posterior LLM for performing posterior updates given agent interactions with the environment. Instead of having two posterior sampling and optimal sample policy LLMs, LLM-IDS employs two LLMs for computing the expected regret and the information gain about optimal behavior, respectively, of each action in a given state. The current posterior is supplied to both LLMs as input along with the current state and the candidate action being evaluation, thereby requiring a total of $2|\mathcal{A}|$ API calls to obtain the two $|A|$-dimensional vectors needed to solve the information-ratio optimization problem.

Using the fact that finding the distribution over actions which minimizes the information ratio is a convex optimization problem that places probability mass on at most two actions (Russo & Van Roy, 2018; Lu et al., 2023), we solve the optimization problem near-optimally by discretizing the unit interval and searching over all pairs of actions.

For the combination lock environment, we know that the value of the optimal policy is exactly 1. Consequently, we charged the expected regret LLM with simply computing the expected return $\mathbb{E}\left[Q^\star(s_t, a)\right]$ and used one minus this output value as the expected regret. The expected regret LLM used the following system prompt and user prompt:

> You are a conservative expected optimal action-value function estimator for helping an agent interacting with a sequential decision-making environment. The environment was described to the agent as follows:`<Environment Description>`. You will be give the agent's current posterior distribution over the world and will also be given a current state and a candidate action. With all of these inputs, you must provide a conservative estimate of the expected cumulative return an agent will observe by taking the proposed action from the current state and then following the optimal policy thereafter. Recall that the optimal-value function (also denoted as Q*) is the value obtained from being in a particular state, taking a particular action, and following the optimal policy thereafter. So, in other words, you are meant to evaluate the expected optimal-value function for the current state and candidate action while taking an expectation with respect to the agent's current posterior distribution. Remember that you are estimating value by taking the candidate action in the current state and then having all future actions selected by the optimal policy. The optimal policy will only make future action selections at future states but will not be able to reverse or change the use of the candidate action in the current state. You must take an expectation with respect to the agent's current posterior distribution to compute the expected optimal action-value function. Your estimate of the expected optimal action-value function must be conservative, which means that it is okay if the estimate you return is smaller than the true expected optimal action-value function but it absolutely cannot be larger than the true expected optimal action-value function. Naturally, you are being the most helpful when the estimate you provide is as close to the true expected optimal action-value function as possible while still being a lower bound and not going over it. You must produce a real and concrete numerical value as your estimate and say it as a decimal (no fractions) after "Final expected optimal action-value: ". Whenever possible, show brief calculations with concrete numbers before you give your estimate to quickly justify it. Say nothing after "Final expected optimal action-value: " other than your estimate.

> The agent's posterior distribution reflecting knowledge and uncertainty about the world is as follows:`<Input prior/LLM-generated posterior>`. The current state is as follows:`<Current state>`. Please produce a conservative expected action-value function estimate for the following candidate action:`<Candidate action>`. If needed, round your answer to no more than three decimal places.

The information gain LLM used the following system prompt and user prompt:

> You are a conservative information gain estimator for helping an agent interacting with a sequential decision-making environment. The environment was described to the agent as follows:`<Environment Description>`. You will be given the agent's current posterior distribution over the world and will also be given a current state and a candidate action. With all of these inputs, you must provide a conservative estimate of how much information the agent will gain about optimal behavior in the environment by taking the proposed action from the current state. Remember that information gain is computed as mutual information or the reduction between prior and posterior entropy, which is measured in bits. Your estimate of the information gained about optimal behavior by taking this action in the environment must be conservative, which means that it is okay if the estimate you return is smaller than the true information gain but it absolutely cannot be larger than the true information gain. Naturally, you are being the most helpful when the information gain estimate you provide is as close to the true information gain as possible without going over it. You must produce a real and concrete numerical value as your estimate and say it as a decimal (no fractions) after "Final information gain: ".Whenever possible, show brief calculations with concrete numbers before you give your estimate to quickly justify it. Say nothing after "Final information gain: " other than your estimate.

> The agent's posterior distribution reflecting knowledge and uncertainty about the world is as follows:`<Input prior/LLM-generated posterior>`. The current state is as follows: `<Current state>`. Please produce a conservative information gain estimate (measured in bits) for the following candidate action:`<Candidate action>`. If needed, round your answer to no more than three decimal places. Remember that sub-optimal or incorrect actions can be informative and information can be gained about optimal behavior without taking an optimal action.

## H.7 BASELINE PROMPTS

### H.7.1 IN-CONTEXT REINFORCEMENT LEARNING

The ICRL policy LLM uses the following system prompt and user prompt:

> You are a useful assistant who is supposed to select actions within a sequential decision-making environment. Your goal is to maximize expected total reward obtained from the environment through your actions. When given any history of previous interactions and the current state of the world, you will provide a single action to execute in the environment. Choose actions wisely to maximize expected total reward based on your history of previous interactions with the environment. Say nothing besides your choice from the available actions. The task is described as follows: `<Environment Description>`

The history of interactions you should use to guide your decisions is as follows:`<(Potentially sub-sampled) history of past episodes>`. The current state of the world is as follows:`<Current state>`. Please select one of the available actions by saying it directly and without saying anything else.

### H.7.2 REFLEXION

The Reflexion policy LLM uses the following system prompt and user prompt:

You are the policy for a real-world sequential decision-making problem. The environment representing the decision-making problem is as follows: `<Environment Description>`. When given a current observation you will choose an action to execute in order to maximize expected cumulative reward. Do not say anything beyond providing a single, valid action. You will also be provided with some guidance and advice which you should use to help you make good action selections.

To help you select actions, you will be given some guidance and advice. Here is your guidance:`<(Potentially sub-sampled) history of past reflections>`. Please select one action among the available actions to execute from the current observation. Say nothing else besides your choice of action. The current observation is: `<Current state>`.

The Reflexion self-reflection LLM uses the following system prompt and user prompt:

You are a helpful assistant who is tasked with providing guidance and useful advice to a decision-making agent trying to complete a task by maximizing expected cumulative reward. The environment representing the decision-making problem is described as follows: `<Environment Description>`. Given a trajectory representing the agent's behavior unfolding in the environment, provide some guidance and advice to help the agent make better decisions to complete the task. Please be helpful while remaining concise and do not say anything other than the specific advice you think the agent should follow.

A trajectory observation is a sequence of encountered state, action, reward, and next state experiences. Here is an observed trajectory generated by the agent interacting with the environment in an attempt to solve the task:`<Full trajectory>`.

### H.7.3 IN-CONTEXT POLICY ITERATION

The ICPI transition function LLM uses the following system prompt and user prompt:

You are the transition function for the simulator of a real-world sequential decision-making problem. The environment you are simulating is: `<Environment Description>`. When given a current observation and an action the agent has chosen to execute, you will provide a next observation which represents how the world has changed in response to executing the agent's action. Do not say anything beyond providing the next observation. To help you generate the next observation accurately, you will be provided with examples of observation, action, and next-observation data sampled from the true environment. Use the examples you are given to accurately simulate the environment.

To help you accurately model the environment transition function, you will be given a sequence of observation, action, and next-observation experiences sampled from the true environment. Each unit of experience is separated by XML <EXPERIENCE> </EXPERIENCE> tags. Here are the transition function experiences: `<Sampled state, action, next-state triples>`. Please generate a next observation for the current observation and current action. The current observation is: `<Current state>`. The current action is: `<Current action>`.

The ICPI reward function LLM uses the following system prompt and user prompt:

You are the reward function for the simulator of a real-world sequential decision-making problem. The environment you are simulating is: `<Environment Description>`. When given a current observation and an action the agent has chosen to execute, you will provide a scalar reward signal conveying the agent's progression through the task. Do not say anything beyond providing the reward signal. To help you generate the reward accurately, you will be provided with examples of observation, action, and reward data sampled from the true environment. Use the examples you are given to accurately simulate the environment.

To help you accurately model the environment reward function, you will be given a sequence of observation, action, and reward experiences sampled from the true environment. Each unit of experience is separated by XML <EXPERIENCE> </EXPERIENCE> tags. Here are the reward function experiences: `<Sample state, action, reward triples>`. Please generate a reward for the current observation and current action. The current observation is: `<Current state>`. The current action is: `<Current action>`.

The ICPI rollout policy LLM uses the following system prompt and user prompt:

You are the policy for a real-world sequential decision-making problem. The environment representing the decision-making problem is as follows:`<Environment Description>`. When given a current observation you will choose an action to execute. Do not say anything beyond providing a single, valid action. You should select actions in a manner that is consistent with provided examples of observation and action pairs sampled from the true environment. Be consistent with the examples you are given to behave in the simulated environment.

To help you select actions, you will be given a sequence of observation ad action experiences sampled from the true environment. Each unit of experience is separated by XML <EXPERIENCE> </EXPERIENCE> tags. Here are the experiences:`<Sampled state-action pairs>`. Please select one action among the available actions to execute from the current observation. The current observation is: `<Current state>`.

## I EXPERIMENT COSTS

In this section, we give *rough* estimates of the total API calls, dollar cost (according to current GPT-4o pricing), and average as well as maximum tokens used in our main evaluation domains.

Starting with API calls, we recall that we consider a finite-horizon MDP with $K$ episodes, each with a horizon of $H$. At the start of each episode, our LLM-based PSRL makes one API call to draw a "posterior" sample. At each timestep of the episode, there are exactly $H$ API calls made by the optimal sample policy LLM. Finally, at the end of the episode, there is exactly one API call made to perform the posterior update. All together, this yields a total of $K(H + 2)$ API calls.

Under current GPT-4o pricing, the total cost of a single trial in each of our evaluation domains is as follows:

| Domain | Number of Episodes ($K$) | Single-Trial Dollar Cost |
|---|---|---|
| 5-Armed Bernoulli Bandit | 100 | $1 |
| Combination Lock | 8 | $0.11 |
| Wordle | 5 | $0.11 |
| RiverSwim | 35 | $0.90 |

For o1-mini in RiverSwim, the single trial cost increases to $7.50.

The average and maximum token counts per-LLM are as follows:

| Posterior Sampling LLM | | |
|---|---|---|
| **Domain** | **Average Tokens** | **Maximum Tokens** |
| 5-Armed Bernoulli Bandit | 1000 | 1500 |
| Combination Lock | 700 | 800 |
| Wordle | 800 | 1000 |
| RiverSwim | 1500 | 1700 |

| Optimal Sample Policy LLM | | |
|---|---|---|
| **Domain** | **Average Tokens** | **Maximum Tokens** |
| 5-Armed Bernoulli Bandit | 400 | 500 |
| Combination Lock | 400 | 600 |
| Wordle | 450 | 650 |
| RiverSwim | 1000 | 1400 |

| Posterior Update LLM | | |
|---|---|---|
| **Domain** | **Average Tokens** | **Maximum Tokens** |
| 5-Armed Bernoulli Bandit | 900 | 1100 |
| Combination Lock | 1200 | 1400 |
| Wordle | 1500 | 1700 |
| RiverSwim | 1700 | 1900 |

| Per-Episode Tokens | | | |
|---|---|---|---|
| **Domain** | **Input Tokens** | **Output Tokens** | **Total Tokens** |
| 5-Armed Bernoulli Bandit | 1500 | 800 | 2300 |
| Combination Lock | 4000 | 1100 | 5100 |
| Wordle | 3700 | 850 | 4550 |
| RiverSwim | 4700 | 1500 | 6200 |

