# OpenReview forum: "Toward Efficient Exploration by Large Language Model Agents"
_ICLR.cc/2026/Conference — ICLR 2026 Poster_

### Official Review · Reviewer_hew5 · 2025-10-30

**Soundness:** 4
**Presentation:** 4
**Contribution:** 3
**Rating:** 8
**Confidence:** 4

**Summary:**

This submission studies how large language models can be used to solve text-based decision-making problems under uncertainty. While prior work has largely focused on prompting or fine-tuning language models to solve decision making tasks, this work instead uses LLMs to implement certain aspects of classical RL algorithms.

In particular, the authors show how to use language models to implement the classical Posterior Sampling for Reinforcement Learning (PSRL) algorithm (an extension of Thompson sampling from bandits to the RL setting) to text-based tasks. At a high level, they do so by instantiating PSRL with three LLM oracles: one to sample from a “textual posterior”, one to select actions given the current state that maximize value in a way that is consistent with the posterior sampling LLM, and one to update the PSRL agent’s knowledge and residual uncertainty about the world, akin to an (approximate) posterior update.

The authors empirically evaluate their LLM-based PSRL algorithm on both tabular RL tasks and on text-based tasks on which classic tabular RL algorithms are not applicable. They find that their algorithm outperforms other methods of using LLMs for solving these tasks (particularly when instantiated with the o1-mini reasoning model).

**Strengths:**

The authors’ proposed idea intuitively makes a lot of sense: we should try to leverage all of the great existing work on RL when trying to solve text-based decision-making tasks under uncertainty, instead of coming up with new algorithms. The choice to use PSRL also makes sense, as it is naturally amenable to being instantiated with LLM oracles. From a practical implementation standpoint, it is nice that LLM-based PSRL can be easily upgraded as newer models are released by just “plugging them in” to the algorithm. The paper’s writing was clear, and the experiments were fairly comprehensive (more on this below).

**Weaknesses:**

While there are no major weaknesses, the ideas in this paper are not particularly deep, and the empirical results are not terribly surprising. It makes sense that LLM-based PSRL would out-perform other methods on text-based tasks, although I would have liked to have seen some experiments on a larger task to really show off the power of this method (e.g. a text-based role-playing game like dungeons and dragons). With that being said, I do not believe that these criticisms significantly detract from the benefits of this submission.

**Questions:**

Do you have any thoughts on how one would extend other RL algorithms to textual settings?

---

> ### Author Response · Authors · 2025-11-21
>
> We thank the reviewer for their time and effort in reviewing our paper.
>
>
> > the ideas in this paper are not particularly deep, and the empirical results are not terribly surprising.
>
>
> We agree with the reviewer that empirical results showing an RL algorithm designed with exploration in mind can offer superior exploration capabilities to algorithms that make no concerted effort to engage with exploration isn’t surprising. However, the fact that the latter baseline LLM agents evaluated in our work are so popular (particularly Reflexion) would suggest that the importance of addressing exploration in the design of LLM agents needs to be made more transparent, hence the title of our paper.
>
>
> We view the “deep” idea behind this work as corroborating the notion that RL algorithms can be used as a template for creating LLM agents. This is an idea that potentially has significant impact and scope, given the widespread deployment of LLM agents, and our paper provides a first test of this idea.
>
>
> >  I would have liked to have seen some experiments on a larger task to really show off the power of this method
>
>
> The primary goal of our paper is to highlight the foundational concepts relevant to building LLM-based RL algorithms and to offer the didactic sanity checks which justify those sorts of grander experiments in future work. This is a first step towards being able to run such algorithms at scale – something that is a challenge given current computing resources, as with other work on LLM agents. The evaluations that we perform are comparable to other evaluations of frameworks for designing LLM agents in the literature, and show that our approach outperforms those frameworks.
>
>
> Note that most benchmarks (both for LLM agents as well as those earlier ones developed for deep RL) often bring a myriad of sequential decision-making challenges (partial observability, exploration, generalization, transfer, non-stationarity, etc.) together thereby creating a conflated signal in the evaluation. If an agent fails to perform well relative to others, it is quite challenging to perform credit assignment and determine which challenge(s) is the culpable root cause. In contrast, each one of our evaluation domains represents a problem where the only obstacle to data efficiency is exploration. Consequently, these tasks represent unambiguous (arguably, quintessential) problems where a lack of success can clearly be ascribed to a failure of effective exploration. It then follows that improved results in each of these domains provides a clear signal that some form of progress on the axis of exploration has been made, without erroneously (over-)claiming success along orthogonal axes.
>
>
> From a more practical perspective, one reason to not initially focus on larger-scale tasks are the considerable costs that come from running a thorough empirical evaluation in such domains. While, as evidenced by our Wordle results, scalability in the size of the state space is not of great concern, scalability in the problem horizon is something we candidly did not study so as to keep our financial costs down. We adopt a forward-looking perspective, recognizing these costs are dynamic and will continue to fluctuate as newer, more-capable models (of tomorrow) are made available and older, (soon to be) less-capable models (of today) are priced down. Therefore, these cost barriers are likely transient and should dissipate over time – the fundamental ideas that we propose are likely to continue to be useful in the future as these costs come down. Part of the goal of doing research is to identify these kinds of opportunities over a longer horizon.
>
>
> > Do you have any thoughts on how one would extend other RL algorithms to textual settings?
>
>
> This is an excellent question and certainly a promising avenue for future work. At the moment, we lack a cohesive recipe that we would be prepared to tout as a “catch-all” methodology for “textualizing” RL algorithms. As discussed in the Related Work section (Appendix A), some prior attempts for doing so lack an entirely sensible narrative that warrants textualization in the first place; the UCB agent of Nie et al., (2024) essentially requires a LLM to perform a series of pairwise additions and an argmax operation while remaining limited to environments where classic UCB can already be executed without any LLM intervention.
>
>
> More broadly, we suspect that textualization of other RL algorithms will likely benefit from a willingness to compromise and embrace approximation (in our paper, this was the sacrifice of the true Bayesian posterior over MDPs to a verbal “posterior” — L205-215) as well as clear pieces of functionality that can be anticipated as being realized just as (if not more) effectively in natural language than in another modality.

---

> > ### Comment · Reviewer_hew5 · 2025-11-26
> >
> > Thanks for your reply - I maintain my positive view of this submission and will keep my score.

---

### Official Review · Reviewer_1o6n · 2025-10-31

**Soundness:** 4
**Presentation:** 3
**Contribution:** 3
**Rating:** 6
**Confidence:** 3

**Summary:**

This paper introduces an implementation of PSRL (Posterior Sampling for RL) by using LLMs to approximate each step (posterior sampling, optimal policy rollout, and posterior update) purely via prompting. Extensive experiments on a variety of small-scale tasks (e.g Bernoulli bandit, Wordle) show the effectiveness of LLMs to minimize cumulative regret across increasing horizon lengths, outperforming related works. This work highlights the effectiveness of using LLMs to implement traditional RL algorithms that have been well-studied for their performance benefits.

**Strengths:**

- Paper is well motivated and reasoning is sound.
- Mathematical formulation is clear and alternatives (IDS) are explained and tested
- Strong results on a variety of tasks, and performance improves with stronger LLMs.

**Weaknesses:**

- Generalization to harder tasks is unclear. The formulation depends on using the LLM as an approximate optimal policy, but as we have seen LLMs are often not the optimal policy. Thus it is unclear, if the LLM is especially weak at the given task how well would this method work?
- Significance of contribution. The authors advocate for implementing existing RL algorithms (like PSRL) via LLMs. However, this comes with the overhead of prompt engineering (+ some assumptions as mentioned in bullet point 1) that may make it harder to scale to new environments/tasks.
- Effectiveness of exploration for RL training. PSRL is a well-studied exploration algorithm, however current work in LLMs + RL focus on RL finetuning LLMs mainly using policy-gradient methods. Are the trajectories generated by the proposed method useful to training LLMs with RL on the task (e.g. faster convergence, higher performance ceiling)?

**Questions:**

1) How well does the algorithm perform if the base LLM is especially weak at the given task?
2) Are the trajectories generated by the proposed method useful to training LLMs with RL on the task (e.g. faster convergence, higher performance ceiling)?

---

> ### Author Response · Authors · 2025-11-21
>
> We thank the reviewer for their time and effort in reviewing our paper.
>
> > Extensive experiments on a variety of small-scale tasks (e.g Bernoulli bandit, Wordle)
>
>
>
> While the domains we evaluate our approach in are discrete – reflecting our focus on settings described by language – characterizing these domains as “small-scale tasks” is an oversimplification. Wordle is considered a non-toy evaluation domain for LLM agents (Klissarov et al, 2024, Tajwar et al, 2025) with a considerable degree of complexity (Lokshtanov & Subercaseaux, 2022) behind it. The MDP representing the game of Wordle (we’ll use this as a primary example though analogues of the subsequent exposition will hold for the combination lock and customer service environments as well) has an action space consisting of all 26 letters of the English alphabet and a state space representing the sequence of letters entered by the agent thus far along with a ternary correctness signal per letter (indicating if the letter is correctly placed, correct but incorrectly placed, or not in the target word at all). Putting aside the ternary feedback signals for a moment, this would imply the Wordle MDP already has at least 7,893,600 unique states (all possible 5-letter sequences without repeating letters). Tacking on the multiplier for all possible 243 length-5 ternary feedback sequences yields a total of 1,918,144,800 states. If this is a small-scale task, then all the games in the Atari suite would likely also be small-scale tasks (only 16,777,216 distinct RGB pixel values for all 84 x 84 images yields just under 118.5 billion states, about 50 times the state space of Wordle, and 18 actions).
>
>
> We agree with the reviewer that large-scale empirical demonstrations and new state-of-the-art results on benchmark tasks always make for attractive “eye-catchers” in machine learning papers. The primary goal of our paper is to highlight the foundational concepts relevant to building LLM-based RL algorithms and to offer the didactic sanity checks which justify those sorts of grander experiments in future work. This is a first step towards being able to run such algorithms at scale – something that is a challenge given current computing resources, as with other work on LLM agents. The evaluations that we perform are comparable to other evaluations of frameworks for designing LLM agents in the literature, and show that our approach outperforms those frameworks.
>
>
> Note that most benchmarks (both for LLM agents as well as those earlier ones developed for deep RL) often bring a myriad of sequential decision-making challenges (partial observability, exploration, generalization, transfer, non-stationarity, etc.) together thereby creating a conflated signal in the evaluation. If an agent fails to perform well relative to others, it is quite challenging to perform credit assignment and determine which challenge(s) is the culpable root cause. In contrast, each one of our evaluation domains represents a problem where the only obstacle to data efficiency is exploration. Consequently, these tasks represent unambiguous (arguably, quintessential) problems where a lack of success can clearly be ascribed to a failure of effective exploration. It then follows that improved results in each of these domains provides a clear signal that some form of progress on the axis of exploration has been made, without erroneously (over-)claiming success along orthogonal axes.
>
>
> From a more practical perspective, one reason to not initially focus on larger-scale tasks are the considerable costs that come from running a thorough empirical evaluation in such domains. While, as evidenced by our Wordle results, scalability in the size of the state space is not of great concern, scalability in the problem horizon is something we candidly did not study so as to keep our financial costs down. We adopt a forward-looking perspective, recognizing these costs are dynamic and will continue to fluctuate as newer, more-capable models (of tomorrow) are made available and older, (soon to be) less-capable models (of today) are priced down. Therefore, these cost barriers are likely transient and should dissipate over time – the fundamental ideas that we propose are likely to continue to be useful in the future as these costs come down. Part of the goal of doing research is to identify these kinds of opportunities over a longer horizon.

---

> > ### Author Response · Authors · 2025-11-21
> >
> > > Generalization to harder tasks is unclear. The formulation depends on using the LLM as an approximate optimal policy, but as we have seen LLMs are often not the optimal policy.
> >
> >
> > The reviewer is correct that we cannot guarantee that the optimal posterior sample LLM will indeed yield the true optimal behavior for the “posterior” sample. As the reviewer further correctly points out, the LLM need not be the optimal policy but only approximately optimal (as discussed in L242-245). We would further underscore that, even when a sub-optimal policy is executed for the drawn “posterior” sampling in a given episode, the subsequent “posterior” update LLM is never made aware of the “posterior” sample in a given episode; that is, “posterior” updates only occur based on the observed environment trajectory and the transition/reward information contained within. Thus, practically speaking, errors in the optimal policy LLM need not necessarily propagate and entirely stall exploration of the overall LLM-based PSRL agent.
> >
> >
> > While we use PSRL to provide a framework for defining our LLM-based agent, there is no hard requirement that the agent perfectly implement the PSRL algorithm. The key question is whether using this framework results in better empirical performance than other approaches for designing LLM agents that are supposed to engage in exploration, and our results demonstrate that this is the case.
> >
> >
> > >How well does the algorithm perform if the base LLM is especially weak at the given task?
> >
> > To put it simply, the algorithm might perform quite poorly. Our paper advocates for an agent-design principle wherein existing RL algorithms are (approximately) implemented via LLM components. Naturally, if one or more of the requisite pieces of functionality needed to implement the algorithm cannot be successfully realized for a given domain, then it likely follows that the algorithm will have limited chance of either success or delivering on the promised capabilities of the original algorithm (in our case, efficient exploration).
> >
> >
> > We maintain that this reality is actually more of a feature than it is a bug. As discussed in Appendix G (L1299-1310), the ability to successfully achieve the desired functionality in each of the three LLMs should serve as “unit tests” signaling to an agent designer whether or not it is sensible to run our proposed PSRL agent. In this light, the early negative results for RiverSwim reported in our paper for GPT-4o (Appendix C) serve as an illustration of what can go wrong when one or more of these three prerequisite LLMs are not satisfactory in their respective roles. Practically speaking, this foresight may translate into significant savings where time, money, and computational resources aren’t wasted running an agent design that is likely doomed to fail. Additionally, with the transparency of precisely which exploration strategy our proposed LLM-based PSRL agent strives to embody, an agent designer may use their intimate knowledge about a domain of interest to recognize that PSRL would be a poor choice for handling exploration; our informative action bandit (Appendix F) is one such example.
> >
> > > this comes with the overhead of prompt engineering (+ some assumptions as mentioned in bullet point 1) that may make it harder to scale to new environments/tasks.
> >
> > Critiques about the overhead of  “prompt engineering” are equally applicable to most (if not all) other work on LLM agents. Prompts are to LLM agents what hyperparameters are to deep RL agents and all prompts must be engineered to some extent just as all hyperparameters must be tuned to some extent. This does not limit the impact of such contributions – for example, Reflexion (which involves use of a specific set of prompts) has almost 3000 citations. The key insight is in structuring a particular agent design to expose specific prompts/hyperparameters that, in the case of this paper, can yield substantial performance improvements on hard-exploration problems where baseline methods fail to deliver comparable results. More importantly, the prompt engineering involved in this work never embeds a drive to explore or ever outlines the PSRL exploration strategy; prompt engineering is done to transparently deliver on the requisite atomic functions that, when orchestrated together per the PSRL algorithm, organically give rise to efficient exploration.

---

> > > ### Author Response · Authors · 2025-11-21
> > >
> > > > current work in LLMs + RL focus on RL finetuning LLMs mainly using policy-gradient methods.
> > > > Are the trajectories generated by the proposed method useful to training LLMs with RL on the task (e.g. faster convergence, higher performance ceiling)?
> > >
> > >
> > > RL fine-tuning of LLMs represents just one way to work at the intersection of LLMs and RL; while it may indeed dominate the lion’s share of interest in the space, it is still but one non-exhaustive area. Our work sits in an entirely different area within the LLMs + RL intersection.
> > >
> > >
> > > That said, as the reviewer correctly notes, the different area of focus in this paper need not be orthogonal to RL fine-tuning. The genuine answer to the reviewer’s question is simply that we do not know. This question, while certainly interesting, stands unrelated to the questions surrounding exploration efficacy of LLM agents studied in this work. As discussed in L912-921, we can only speculate that to the extent RL fine-tuning is plagued by an exploration challenge in response space (that is, a good response must be actively sought out before any derived feedback signals on it can be internalized and reinforced via fine-tuning), there may be exciting opportunities for future work to generate RL fine-tuning data with our proposed LLM-based PSRL agent that potentially yield significant performance improvements or cost-savings. We would encourage the reviewer to see the work of Dwaracherla et al., (2024) cited in our paper for more insights on exploration specifically as it pertains to RL fine-tuning of LLMs.

---

### Official Review · Reviewer_sqCQ · 2025-11-01

**Soundness:** 2
**Presentation:** 2
**Contribution:** 2
**Rating:** 2
**Confidence:** 5

**Summary:**

This paper proposes using large language models (LLMs) to implement a well-established reinforcement learning (RL) algorithm of Posterior Sampling for Reinforcement Learning (PSRL). They show through experiments in bandit, tabular, and natural-language MDP environments that this “LLM-based PSRL” framework achieves more data-efficient exploration than other agent designs.

1. While paper does present Regret plots, it omits statistical significance tests. Can the authors please report those?
2. From my point of view, the paper has limited novelty. The core contribution is an application of PSRL through LLMs rather than a theoretical or methodological advance in RL or LLM architectures. Please let me know if I have misunderstood or didn't fully notice another novelty presented.
3. I understand that the authors are trying to present more transparent results to understand the importance of PSRL. But, to me, the experiments remain toy and fall short of demonstrating scalability or robustness in realistic open-ended tasks. Maybe the authors could consider adding more realistic domains? Embodied AI could be an option, maybe?
4. I am not convinced that the posterior that the uncertainty the authors refer to is actually calibrated. There’s no formal analysis of whether LLM-generated samples reflect valid uncertainty quantification or just linguistic variability.
5. There are many LLMs used in this paper. I would like to ask the authors to show an efficiency comparison (e.g., token or cost per reward improvement), especially with the models that are used.
6. There are some claims that I am not sure I understand: 'one might hope to see an LLM-based implementation of PSRL exhibit similar robustness in practice' - I am not sure what this means. This is not substantiated very well in the paper. There are no new regret bounds provided within the non-convex world of LLMs. So, I am not sure how to understand such claims.  Like, the LLM introduces stochasticity and additional approximations. So do the theoretical results still hold?

**Strengths:**

Please see above

**Weaknesses:**

Please see above

**Questions:**

Please see above

---

> ### Author Response · Authors · 2025-11-21
>
> We thank the reviewer for their time and effort in reviewing our paper.
>
> > While paper does present Regret plots, it omits statistical significance tests. Can the authors please report those?
>
>
> We are unsure of what “statistical significance tests” the reviewer believes we have omitted. As is standard, we have reported cumulative regret curves with one standard error visualized.
>
>
> To address the reviewer’s concern, however, that the conventional reporting of our findings is not statistically significant, we use one-sided unpaired permutation tests to evaluate the hypothesis that our proposed LLM-based PSRL agent achieves lower final cumulative regret (that is, the cumulative regret obtained at the end of all $K$ episodes). For each task, we conducted pairwise comparisons between our approach and the baseline LLM agents, applying the Holm-Bonferroni method to control the family-wise error rate at $\alpha = 0.05$. Across all tasks, we reaffirm statistically-significant improvements in final cumulative regret by our LLM-based PSRL agent over our strongest LLM agent baselines (Reflexion and ICRL). We also report effect sizes — defined as the difference in mean cumulative regret (baseline minus ours) — with 95% bootstrap confidence intervals computed by resampling runs independently per algorithm.
>
> | Task                    | Baseline              | Effect Size (Δ = mean(baseline) − mean(ours)) | 95% CI         | Holm-adjusted p-value |
> |-------------------------|-----------------------|-----------------------------------------------|----------------|------------------------|
> | Customer Service Bandit | ICRL ($p = 1.0$)      | 5.4                                           | [1.85, 8.85]   | 0.002                  |
> |                         | Reflexion             | 11.8                                          | [7.20, 15.85]  | 0.002                  |
> | RiverSwim               | ICRL ($p = 1.0$)      | 6.03                                          | [2.25, 9.65]   | 0.011                  |
> |                         | Reflexion             | 12.08                                         | [9.97, 14.16]  | 0.011                  |
> | CombinationLock         | ICRL ($p = 0.1$)      | 4.0                                           | [3.39, 4.65]   | 0.0002                 |
> |                         | ICRL ($p = 0.5$)      | 3.45                                          | [2.59, 4.25]   | 0.0004                 |
> |                         | ICRL ($p = 1.0$)      | 2.15                                          | [1.25, 3.05]   | 0.0006                 |
> |                         | Reflexion             | 2.35                                          | [1.39, 3.25]   | 0.0008                 |
> | Wordle                  | ICRL ($p = 1.0$)      | 2.18                                          | [1.63, 2.72]   | 0.0002                 |
> |                         | Reflexion             | 2.50                                          | [1.90, 3.07]   | 0.0004                 |
>
>
>
> > the paper has limited novelty. The core contribution is an application of PSRL through LLMs rather than a theoretical or methodological advance in RL or LLM architectures.
>
>
> We view the novel contribution of our paper to be the proposal that LLM agent designs can be created from existing RL algorithms. We don’t claim to be the creators of those RL algorithms, or the first to highlight exploration as a key challenge for data-efficient RL, or the inventors of LLM agents. The confluence of these three concepts, however, is the novel contribution that our paper offers which, to the best of our knowledge, has not appeared in the prior literature on LLM agents. Thus, our paper offers a significant methodological advance in how researchers approach the design of LLM agents for hard-exploration, natural-language environments.

---

> > ### Author Response · Authors · 2025-11-21
> >
> > > the experiments remain toy and fall short of demonstrating scalability or robustness in realistic open-ended tasks. Maybe the authors could consider adding more realistic domains? Embodied AI could be an option, maybe?
> >
> >
> > While the domains we evaluate our approach in are discrete – reflecting our focus on settings described by language – characterizing these domains as “toy” is an oversimplification. Wordle is considered a non-toy evaluation domain for LLM agents (Klissarov et al, 2024, Tajwar et al, 2025) with a considerable degree of complexity (Lokshtanov & Subercaseaux, 2022) behind it. The MDP representing the game of Wordle (we’ll use this as a primary example though analogues of the subsequent exposition will hold for the combination lock and customer service environments as well) has an action space consisting of all 26 letters of the English alphabet and a state space representing the sequence of letters entered by the agent thus far along with a ternary correctness signal per letter (indicating if the letter is correctly placed, correct but incorrectly placed, or not in the target word at all). Putting aside the ternary feedback signals for a moment, this would imply the Wordle MDP already has at least 7,893,600 unique states (all possible 5-letter sequences without repeating letters). Tacking on the multiplier for all possible 243 length-5 ternary feedback sequences yields a total of 1,918,144,800 states. If this is a toy environment then all the games in the Atari suite would likely also be toy environments (only 16,777,216 distinct RGB pixel values for all 84 x 84 images yields just under 118.5 billion states, about 50 times the state space of Wordle, and 18 actions).
> >
> >
> >
> > We agree with the reviewer that large-scale empirical demonstrations and new state-of-the-art results on benchmark tasks always make for attractive “eye-catchers” in machine learning papers. The primary goal of our paper is to highlight the foundational concepts relevant to building LLM-based RL algorithms and to offer the didactic sanity checks which justify those sorts of grander experiments in future work. This is a first step towards being able to run such algorithms at scale – something that is a challenge given current computing resources, as with other work on LLM agents. The evaluations that we perform are comparable to other evaluations of frameworks for designing LLM agents in the literature, and show that our approach outperforms those frameworks.
> >
> >
> > Note that most benchmarks (both for LLM agents as well as those earlier ones developed for deep RL) often bring a myriad of sequential decision-making challenges (partial observability, exploration, generalization, transfer, non-stationarity, etc.) together thereby creating a conflated signal in the evaluation. If an agent fails to perform well relative to others, it is quite challenging to perform credit assignment and determine which challenge(s) is the culpable root cause. In contrast, each one of our evaluation domains represents a problem where the only obstacle to data efficiency is exploration. Consequently, these tasks represent unambiguous (arguably, quintessential) problems where a lack of success can clearly be ascribed to a failure of effective exploration. It then follows that improved results in each of these domains provides a clear signal that some form of progress on the axis of exploration has been made, without erroneously (over-)claiming success along orthogonal axes.
> >
> >
> > From a more practical perspective, one reason to not initially focus on larger-scale tasks are the considerable costs that come from running a thorough empirical evaluation in such domains. While, as evidenced by our Wordle results, scalability in the size of the state space is not of great concern, scalability in the problem horizon is something we candidly did not study so as to keep our financial costs down. We adopt a forward-looking perspective, recognizing these costs are dynamic and will continue to fluctuate as newer, more-capable models (of tomorrow) are made available and older, (soon to be) less-capable models (of today) are priced down. Therefore, these cost barriers are likely transient and should dissipate over time – the fundamental ideas that we propose are likely to continue to be useful in the future as these costs come down. Part of the goal of doing research is to identify these kinds of opportunities over a longer horizon.

---

> > > ### Author Response · Authors · 2025-11-21
> > >
> > > > I am not convinced that the posterior that the uncertainty the authors refer to is actually calibrated. There’s no formal analysis of whether LLM-generated samples reflect valid uncertainty quantification or just linguistic variability.
> > >
> > >
> > >
> > > We note that any use of the word “posterior” in our LLM-based PSRL is not meant to refer to the statistical object that is the true Bayesian posterior over the underlying MDP given the history of interaction thus far (L205-221). We make no claim or assumption of theoretical equivalence whatsoever. We have no reason to suspect that the stochasticity of LLM responses will align well with true sampling of the Bayesian posterior. However, our empirical results clearly demonstrate that, for some environments, sampling LLMs to perform the requisite functions needed by PSRL yields an agent that explores effectively.
> > >
> > > >  I would like to ask the authors to show an efficiency comparison (e.g., token or cost per reward improvement), especially with the models that are used.
> > >
> > >
> > > Per the reviewer’s request, we have now added Appendix G as an updated revision to the paper, which contains token efficiency comparisons across Reflexion, ICRL, and our LLM-based PSRL all under GPT-4o.
> > >
> > >
> > > > 'one might hope to see an LLM-based implementation of PSRL exhibit similar robustness in practice' - I am not sure what this means. This is not substantiated very well in the paper.
> > >
> > >
> > > The full sentence from which that except originates read as “Even when this policy is only approximately-optimal with respect to the posterior sample in a given episode, classic PSRL still admits a Bayesian regret bound (see Section 5.4 of Osband (2016a)) and one might hope to see an LLM-based implementation of PSRL exhibit similar robustness in practice.” In this sentence, we focus on a natural point of concern readers may have about our approach: the policy deployed by the optimal policy LLM is not in fact optimal for the drawn “posterior” sample. To help assuage this concern, we point to an existing theoretical result established for PSRL which yields a Bayesian regret bound even when the policy deployed in each episode is only approximately optimal with respect to the posterior sample, rather than exactly optimal. In the latter part of the sentence (quoted by the reviewer), we simply point out the aspiration that this theory translates well to our LLM-based PSRL agent in practice. Obviously, there is no such guarantee that this will in fact be the case across all environments, which is why it was not firmly stated as a claim or contribution and, therefore, did not warrant substantiation in the paper. We will update this sentence to communicate this more clearly.
> > >
> > > > There are no new regret bounds provided within the non-convex world of LLMs. So, I am not sure how to understand such claims. Like, the LLM introduces stochasticity and additional approximations. So do the theoretical results still hold?
> > >
> > >
> > >
> > > It is true that our work introduces no new regret bounds as it is a purely empirical paper that builds on an existing body of theoretical results. Theoretical guarantees for PSRL are already well-established in the literature (L144-150) and need no further litigation. Analogous proofs of statistical efficiency for our LLM-based PSRL would probably not be possible without a myriad of technical assumptions from which a guarantee would essentially fall out “for free” in a likely uninteresting fashion and without delivering much useful mathematical insight or technical novelty; that is, it would most likely not hold up as a valuable technical contribution for the RL theory community. The point of this paper is to provide a clear, unambiguous demonstration of how LLMs can serve as a contemporary vehicle for lifting principled ideas previously bound only to simpler settings and realizing the benefits of those ideas in natural-language tasks. PSRL is the concrete instantiation of this higher-level principle for agent design studied by this work.

---

### Official Review · Reviewer_s3NC · 2025-11-01

**Soundness:** 3
**Presentation:** 4
**Contribution:** 2
**Rating:** 4
**Confidence:** 4

**Summary:**

This paper proposes a new LLM-based algorithm designed to address the exploration inefficiency observed in existing LLM-agent frameworks. The authors adapt the classical Posterior Sampling for Reinforcement Learning (PSRL) algorithm, which is well known for its exploration efficiency, by replacing its three key subroutines with LLM API calls: (1) posterior sampling that imitates Bayesian (Thompson) sampling, (2) deriving and executing an optimal policy under the sampled model, and (3) updating the posterior for the next episode. The approach is compared against the classical PSRL formulation and other LLM-based agents such as Reflexion and ICRL. Across several simple, mostly tabular environments (e.g., Wordle, RiverSwim, and bandits), the LLM-PSRL agent achieves lower cumulative regret, attributed to more directed exploration. Empirically, the work demonstrates that an LLM can approximate Bayesian reasoning behavior through structured prompting and temperature-controlled stochasticity.

**Strengths:**

The paper is clearly written and transparent about its limitations, providing sufficient implementation details to reproduce the results. The structure and presentation are excellent, and each component of the algorithm and experiment is explained in a straightforward and understandable manner. Conceptually, the idea of embedding a classical reinforcement learning framework (PSRL) within an LLM-driven architecture is both interesting and original. It demonstrates that language models can approximate Bayesian reasoning through prompting alone. This makes the paper an insightful contribution for researchers exploring how LLMs can emulate principled decision-making methods.

**Weaknesses:**

The main limitation of the paper lies in its scope and the strength of its claims. While the proposed method demonstrates improved exploration efficiency, (1) its scalability to complex or real-world RL settings remains unclear. (2) The paper assumes that LLM agents inherit the exploration inefficiency of classical RL, *but this assumption is not empirically established and may not hold in the same way for language-based agents.* Consequently, the core motivation of addressing “poor exploration” in LLMs feels somewhat speculative. In addition, the decision to restrict comparisons to the LLM-agent design space (Reflexion and ICRL) limits the broader impact of the results. Including more diverse or recent LLM-agent baselines, once available, would make the contribution more convincing. Overall, the novelty is interesting, but the empirical and conceptual scope remains narrow.

**Questions:**

1. You assume that LLM-based agents, like classical RL agents, suffer from poor exploration leading to data inefficiency. Could you provide stronger justification or empirical evidence supporting this assumption? How do you rule out other possible causes such as context limitations?
2. Have you verified that the LLM’s stochasticity aligns with true Bayesian sampling behavior, or is this assumption based on temperature-induced randomness alone?
3. Is your method scalable to higher-dimensional or continuous RL environments? If not, what are the main bottlenecks (e.g., token cost, prompt size, sampling variability)?
4. You mention that practical PSRL implementations beyond tabular MDPs face computational hurdles, yet your experiments are conducted only on tabular-like environments. Could you clarify why you chose not to include a non-tabular or higher-dimensional benchmark?

---

> ### Author Response · Authors · 2025-11-21
>
> We thank the reviewer for their time and effort in reviewing our paper.
> > Across several simple, mostly tabular environments (e.g., Wordle, RiverSwim, and bandits)
> > your experiments are conducted only on tabular-like environments. Could you clarify why you chose not to include a non-tabular or higher-dimensional benchmark?
>
>
>
> While the domains we evaluate our approach in are discrete – reflecting our focus on settings described by language – characterizing these domains as “tabular-like environments” is an oversimplification. A tabular environment refers to a sequential decision-making problem that meets two specific criteria: (1) it has a finite state-action space and (2) the cardinality of the state-action space is sufficiently small such that salient components of a RL algorithm (policy, value function, or estimates of the transition and reward functions) may be represented as tables or arrays.
>
>
>
> Wordle is considered a non-toy evaluation domain for LLM agents (Klissarov et al, 2024, Tajwar et al, 2025) with a considerable degree of complexity (Lokshtanov & Subercaseaux, 2022) behind it. The MDP representing the game of Wordle (we’ll use this as a primary example though analogues of the subsequent exposition will hold for the combination lock and customer service environments as well) has an action space consisting of all 26 letters of the English alphabet and a state space representing the sequence of letters entered by the agent thus far along with a ternary correctness signal per letter (indicating if the letter is correctly placed, correct but incorrectly placed, or not in the target word at all). Putting aside the ternary feedback signals for a moment, this would imply the Wordle MDP already has at least 7,893,600 unique states (all possible 5-letter sequences without repeating letters). Tacking on the multiplier for all possible 243 length-5 ternary feedback sequences yields a total of 1,918,144,800 states. If this is a tabular MDP, then all the games in the Atari suite would likely also be tabular MDPs (only 16,777,216 distinct RGB pixel values for all 84 x 84 images yields just under 118.5 billion states, about 50 times the state space of Wordle, and 18 actions). These domains thus push the limit of what could reasonably be characterized as tabular environments, and are more comparable to other challenging RL problems.
>
>
>
> For the response regarding a “higher-dimensional benchmark,” please see the response below.

---

> > ### Author Response · Authors · 2025-11-21
> >
> > > its scalability to complex or real-world RL settings remains unclear.
> >
> > The primary goal of our paper is to highlight the foundational concepts relevant to building LLM-based RL algorithms and to offer the didactic sanity checks which justify those sorts of grander experiments in future work. This is a first step towards being able to run such algorithms at scale – something that is a challenge given current computing resources, as with other work on LLM agents. The evaluations that we perform are comparable to other evaluations of frameworks for designing LLM agents in the literature, and show that our approach outperforms those frameworks.
> >
> >
> > Note that most benchmarks (both for LLM agents as well as those earlier ones developed for deep RL) often bring a myriad of sequential decision-making challenges (partial observability, exploration, generalization, transfer, non-stationarity, etc.) together thereby creating a conflated signal in the evaluation. If an agent fails to perform well relative to others, it is quite challenging to perform credit assignment and determine which challenge(s) is the culpable root cause. In contrast, each one of our evaluation domains represents a problem where the only obstacle to data efficiency is exploration. Consequently, these tasks represent unambiguous (arguably, quintessential) problems where a lack of success can clearly be ascribed to a failure of effective exploration. It then follows that improved results in each of these domains provides a clear signal that some form of progress on the axis of exploration has been made, without erroneously (over-)claiming success along orthogonal axes.
> >
> >
> > From a more practical perspective, one reason to not initially focus on larger-scale tasks are the considerable costs that come from running a thorough empirical evaluation in such domains. While, as evidenced by our Wordle results, scalability in the size of the state space is not of great concern, scalability in the problem horizon is something we candidly did not study so as to keep our financial costs down. We adopt a forward-looking perspective, recognizing these costs are dynamic and will continue to fluctuate as newer, more-capable models (of tomorrow) are made available and older, (soon to be) less-capable models (of today) are priced down. Therefore, these cost barriers are likely transient and should dissipate over time – the fundamental ideas that we propose are likely to continue to be useful in the future as these costs come down. Part of the goal of doing research is to identify these kinds of opportunities over a longer horizon.
> >
> > > The paper assumes that LLM agents inherit the exploration inefficiency of classical RL, but this assumption is not empirically established and may not hold in the same way for language-based agents.
> > > the core motivation of addressing “poor exploration” in LLMs feels somewhat speculative.
> > > You assume that LLM-based agents, like classical RL agents, suffer from poor exploration leading to data inefficiency. Could you provide stronger justification or empirical evidence supporting this assumption? How do you rule out other possible causes such as context limitations?
> >
> >
> > We believe there is widespread agreement on the fundamental need for LLM *agents* (not necessarily the constituent LLMs themselves) to engage in the exploration-exploitation trade-off. That exploration is a fundamental obstacle to achieving data-efficient RL is an objective, well-established fact (Sutton & Barto; 1998). Consequently, LLM agents (a subset of RL agents) engaging with sequential decision-making problems under uncertainty to synthesize optimal behavior from trial-and-error feedback over time must also confront the exploration challenge. Of course, moving away from the general RL problem, there are environments which do not pose a significant exploration challenge (analogous to those deep RL environments where naive $\epsilon$-greedy exploration or entropy regularization suffice); naturally, our paper and empirical evaluation are not focused on such decision-making problems.
> >
> >
> > Beyond appealing to the logic outlined above, the reviewer is further incorrect about the lack of existing work establishing how LLM agents struggle to contend with the exploration challenge; our paper already cites notable examples (Krishnamurthy et al.; 2024, Nie et al.; 2024).
> >
> > Finally, discounting the logic and existing work discussed above, the empirical evaluation of our paper assesses popular LLM agent baselines in hard-exploration problems, where they demonstrably fail to achieve more data-efficient learning than our proposed LLM-based PSRL agent. Any subsequent, internal diagnoses of why these difficulties emerge within any particular agent design (for instance, due to context limitations) cannot be substituted for the actual reason for failure in the task itself (namely, poor and inefficient exploration).

---

> > > ### Author Response · Authors · 2025-11-21
> > >
> > > > the decision to restrict comparisons to the LLM-agent design space (Reflexion and ICRL) limits the broader impact of the results. Including more diverse or recent LLM-agent baselines, once available, would make the contribution more convincing
> > >
> > >
> > > Our proposed approach is intended to replace ad hoc designs for agent architectures. Consequently, comparison against LLM-agent design strategies is the appropriate source for baselines. We are unsure what other LLM-agent design approaches the reviewer has in mind – we also included more recent work that has taken approaches inspired by RL and show that our approach performs well in those comparisons. If there are specific evaluations the reviewer would like to see we can try to run those in the remaining time for the rebuttal.
> > >
> > >
> > > > Have you verified that the LLM’s stochasticity aligns with true Bayesian sampling behavior, or is this assumption based on temperature-induced randomness alone?
> > >
> > >
> > > We note that any use of the word “posterior” in our LLM-based PSRL is not meant to refer to the statistical object that is the true Bayesian posterior over the underlying MDP given the history of interaction thus far (L205-221). We make no claim or assumption of theoretical equivalence whatsoever. We have no reason to suspect that the stochasticity of LLM responses will align well with true sampling of the Bayesian posterior. However, our empirical results clearly demonstrate that, for some environments, sampling LLMs to perform the requisite functions needed by PSRL yields an agent that explores effectively.
> > >
> > > > Is your method scalable to higher-dimensional or continuous RL environments? If not, what are the main bottlenecks
> > >
> > >
> > >
> > > We interpret  “higher-dimensional or continuous RL environments” as the reviewer asking about the applicability of our proposed LLM-based PSRL agents to continuous-control environments like Mujoco. No large *language* model agent (neither ours nor baselines) will be particularly amenable to continuous-control problems with vector-valued states and actions. While agents that integrate a deep RL component suitable for handling the high-dimensional states and actions with a LLM for hierarchical abstraction in natural language (for instance, the SayCan agent by Google Robotics) are possible, such hybrid agents are outside the scope of this work.

---

### Author Response · Authors · 2025-12-02
**Summary of Initial Reviews, Rebuttals, & Discussions**

Dear AC,


Thank you for your time and continued efforts to support ICLR 2026. Our condolences for the additional labor that now befalls you in light of recent events. To help ameliorate the burden of your task and assist in synthesizing information salient to a final decision on our paper, we offer the following summary of the initial reviews as well as our responses and corresponding paper updates.



At an abstract level, our paper focuses on a specific LLM agent-design principle that existing RL algorithms can be used as a template for creating LLM agents. Concretely, our paper studies a first instantiation of this principle through a classic RL algorithm called Posterior Sampling for Reinforcement Learning (PSRL), well-known in the literature for its statistically-efficient exploration properties inherited from Thompson Sampling. Intuitively, as LLM agents are a subset of all RL agents and exploration is a core data-efficiency challenge of RL, it follows that the ability for LLM agents to effectively explore is critical to their deployment and adoption. Our primary contribution, by virtue of this design principle, is demonstrating how LLMs can serve as a contemporary vehicle for delivering the efficient exploration of classic PSRL well beyond the confines of tabular MDPs to hard-exploration, natural-language tasks.


Despite high variance in the scores of our paper (8, 6, 4, 2), all reviewers to varying degrees acknowledged our success in delivering on the contributions stated above. Reviewer s3NC noted that “the idea of embedding a classical reinforcement learning framework (PSRL) within an LLM-driven architecture is both interesting and original” thereby making for “an insightful contribution for researchers exploring how LLMs can emulate principled decision-making methods.” Reviewer hew5 noted that our “experiments were fairly comprehensive” evaluating “on both tabular RL tasks and on text-based tasks on which classic tabular RL algorithms are not applicable”, from which Reviewer sqCQ concluded that our LLM-based PSRL agent “achieves more data-efficient exploration than other agent designs” and Reviewer 1o6n noted how our work “highlights the effectiveness of using LLMs to implement traditional RL algorithms that have been well-studied for their performance benefits.”


The main criticism levied by all reviewers was a concern regarding the scalability of our approach. Our response to this concern was twofold. First, to clear up a confusion for some reviewers that our evaluation only studied tabular MDPs, we detailed the non-toy nature of our Wordle task (consisting of a finite state space with just over 1.9 billion states). Secondly, we emphasized that our evaluation was meticulously targeted on providing a didactic assessment of exploration efficacy, whereas large-scale domains often conflate multiple data-efficiency challenges (partial observability, generalization, etc.) and obfuscate the credit assignment to positive improvements and negative results. The foundational concepts and sanity checks studied in this work set the stage for those grander experiments in future work, justifying the investment of time and resources.

Reviewers s3NC and sqCQ erroneously believed our work makes a theoretical equivalence claim or some other formal engagement with the true Bayesian posterior distribution. We clarified that this is absolutely not a claim of our empirical paper and elucidated how such theoretical guarantees are likely to either be not fruitful or possibly redundant given the extent to which PSRL has already largely been an object of theoretical study in the literature. A central advantage of this work is making those theoretical benefits practically realizable at an unprecedented scale.

Reviewer sqCQ gave our paper an initial score of 2. Beyond the two issues resolved above, they requested additional statistical significance tests for our empirical results (beyond the standard reporting of cumulative regret curves with standard error bars computed from no fewer than 20 trials) as well as a token efficiency comparison. The former was given as a table in our rebuttal response whereas the latter was provided in an updated version of the paper as Appendix G. The conclusions from these additional experiments are that (1) our proposed LLM-based PSRL agent does achieve a statistically-significant improvement in final cumulative regret over baseline LLM agents and (2) for a given token budget, our LLM-based PSRL agent displays comparable or superior exploration efficiency, as measured by cumulative regret. We believe that our response has addressed all of the issues that motivated their low initial review score.

---

> ### Author Response · Authors · 2025-12-02
> **Summary of Initial Reviews, Rebuttals, & Discussions (2/2)**
>
> Reviewers s3NC gave our paper an initial score of 4. Beyond the two issues resolved above, they suggested demonstrating scalability through high-dimensional, continuous control tasks, in response to which we pointed out the folly of applying LLMs to tasks with vector-valued state-action spaces. This reviewer also forgot that, in the general RL problem, agents (including LLM agents) must contend with the exploration challenge and, in all likelihood, greatly benefit from an explicit mechanism for doing so. Indeed, our experiments confirm that it is unwise to merely trust a LLM agent to explore by itself and leave it to its own devices, as done by popular agents like Reflexion. Reviewer s3NC hinted at how “more diverse or recent LLM-agent baselines, once available, would make the contribution more convincing,” but failed to either concretely identify a deficiency in our empirical evaluation or make a meaningful suggestion for alternative baseline LLM agents worth consideration. We are unaware of any other salient LLM agent baselines that make a concerted effort to engage with the challenge of exploration without being constrained to simpler problem settings where classic exploration strategies can be run first and then emulated by LLMs (for example, the UCB-style agent of Nie et al., (2024)).
>
> Reviewer 1o6n gave our paper an initial score of 6. Beyond the scalability concern addressed above, the reviewer had concerns regarding prompt engineering and the applicability of our proposed LLM-based PSRL agent if the functionality of one or more constituent LLMs were to be compromised or deficient in some way. To the former, we reminded the reviewer that prompt engineering is an inevitable reality of all LLM agents (just as deep RL agents have numerous hyperparameters to tune with varying degrees of sensitivity). The point of an agent design is to expose particular prompts/hyperparameters that can be appropriately configured to facilitate efficient synthesis of optimal behavior. For the latter concern, we maintain that the outlined failure mode is more of a feature than a bug. The individual functionality of constituent LLMs in our proposed agent serve as “unit tests” for determining whether or not it would be sensible to run our agent. Our preliminary failures with RiverSwim (Appendix C) illustrate the potential negative results from ignoring these warning signs. The inability to provide the three requisite components needed to implement PSRL would suggest that an agent designer would likely be better served by some other LLM agent for solving their task; this is somewhat analogous to observing how a fully-connected MLP is incapable of learning sufficiently-good image features and concluding that it might be a poor choice of model to run on an image classification task.
>
> Finally, Reviewer hew5 gave our paper a score of 8. Beyond the scalability concern addressed above, this reviewer asked about the high-level design principle embodied by our proposed LLM-based PSRL agent and the ability to “textualize” other RL algorithms. We candidly replied that, aside from the two concrete “textualizations” of RL algorithms offered by our work, we are not yet sure of a general-purpose recipe that could guide the RL/LLM agent community on this textualization process. That said, we suspect this is an incredibly promising area for future work whose potential benefits are precisely foreshadowed by the contributions of this paper. We were grateful to hear from Reviewer hew5 post-rebuttal — the sole bit of engagement for our paper during the discussion period — whose positive reply confirmed resolution of the scalability concern and “textualization” question raised in their initial review.
>
> Unfortunately, chaos ensued before we could engage in any further discussions with most of our reviewers. Nevertheless, we thank all the reviewers for their comments and particularly Reviewer hew5 for their engagement with our rebuttal response. We refer the AC to the full discussion below for additional clarifications and discussion.
>
> Best wishes,
>
>
> The Authors

---

### Meta-Review · Area_Chair_FQZt · 2025-12-19

**Summary:**

This paper had a high variance in scores (2,4,6,8). The main criticism raised by all reviewers was that the empirical evaluation focused on overly simple (or even toy) tasks. There were some other criticisms or questions that were raised, such as sensitivity to prompting, statistical significance between the different curves, reporting LLM costs and the like, but I believe these are relatively minor and have adequately been addressed by the authors' response. Several reviewers highlighted the quality of the presentation and soundness of the reasoning (hew5, 1o6n, s3NC). Opinions were mixed regarding the paper’s novelty, with some reviewers describing it as quite original (s3NC) while others finding the contributions incremental (sqCQ).

Given that the primary criticism of the paper was in its choice of tasks for the empirical evaluation (which reviewers found overly simple, although opinions were divided whether this was grounds for rejection or not), and that no new experiments were provided during the rebuttal, I don't think reviewers would have significantly changed their scores.

I share some of the reviewers’ concerns about the simplicity of the tasks. Although the authors point out that some of these tasks have a large state space (similar to Atari), I don't think the size of the state space necessarily reflects the complexity of the task (e.g. adding one dimension of Gaussian noise makes the state space infinite). In any case, in 2025 the Atari benchmark is probably also considered somewhat toy. That being said, I also understand the authors’ point that simple tasks allow careful diagnostics of a particular capability (here exploration).

I do think the core idea of the paper is quite ingenious and could potentially have high impact. One of its key strengths is its simplicity: essentially, you can embed an RL algorithm in the prompt, and get the LLM agent to act in an efficient information-gathering way. Because of this, even though I would have liked a more thorough empirical evaluation, I think it is worth sharing and am recommending **accept**. I expect different researchers can quickly incorporate this idea into their pipelines, which can help with the empirical evaluation aspect.


That being said, I also encourage the authors to keep pushing the idea and proving it out empirically (either to add to this paper for the camera ready, or a future one). I think there are several ways in which the authors could apply their method to more complex benchmarks. Although they rightly point out that applying LLMs to continuous state/action spaces naively will likely not work, there are alternatives — for example, the LLM can coordinate policies which have a natural textual description. Two examples of works which do this are (there are likely others):
- Code As Policies: LLM synthesizes low-level code which combines action primitives into a policy (https://arxiv.org/abs/2209.07753) - has been demonstrated in embodied AI.
- MaestroMotif: LLM composes RL policies with natural language descriptions - has been demonstrated on NetHack (https://arxiv.org/abs/2412.08542)

**Reviewer Concerns:**

Other than the simplicity of the tasks used in the empirical evaluation, most of the reviewer concerns have been answered in a satisfactory manner in my opinion.

**Reviewer Scores:**

sqCQ: 2 -> 2

s3NC: 4 -> 4

1o6n: 6 -> 6

hew5: 8 -> 8

---

### Decision · Program_Chairs · 2026-01-26

Accept (Poster)